# Critical role of the BAF chromatin remodeling complex during murine neural crest development

Kathleen Wung Bi-Lin[1], Pratap Veerabrahma Seshachalam[1], Tran Tuoc[2,3], Anastassia Stoykova[4], Sujoy Ghosh[1], Manvendra K. Singh[1,5]*

1 Program in Cardiovascular and Metabolic Disorders, Duke-NUS Medical School Singapore, Singapore, 2 Department of Human Genetics, Ruhr University of Bochum, Bochum, Germany, 3 Institute of Neuroanatomy, University Medical Center, Georg-August-University Goettingen, Goettingen, Germany, 4 Max-Planck-Institute for Biophysical Chemistry, Goettingen, Germany, 5 National Heart Research Institute Singapore, National Heart Center Singapore, Singapore

* manvendra.singh@duke-nus.edu.sg

**Data Availability Statement:** The authors confirm that all data underlying the findings are fully available without restriction. All relevant data are within the manuscript and its Supporting

## Abstract

The BAF complex plays an important role in the development of a wide range of tissues by modulating gene expression programs at the chromatin level. However, its role in neural crest development has remained unclear. To determine the role of the BAF complex, we deleted BAF155/BAF170, the core subunits required for the assembly, stability, and functions of the BAF complex in neural crest cells (NCCs). Neural crest-specific deletion of *BAF155/BAF170* leads to embryonic lethality due to a wide range of developmental defects including craniofacial, pharyngeal arch artery, and OFT defects. RNAseq and transcription factor enrichment analysis revealed that the BAF complex modulates the expression of multiple signaling pathway genes including Hippo and Notch, essential for the migration, proliferation, and differentiation of the NCCs. Furthermore, we demonstrated that the BAF complex is essential for the Brg1-Yap-Tead-dependent transcription of target genes in NCCs. Together, our results demonstrate an important role of the BAF complex in modulating the gene regulatory network essential for neural crest development.

## Author summary

Neural crest cells (NCCs) are a multipotent and migratory cell population that is induced at the neural plate border during neurulation and contributes to the formation of a wide range of tissues. Defects in the development, differentiation, or migration of NCCs lead to various birth defects including craniofacial and heart anomalies. Here, by genetically deleting BAF155/BAF170, the core subunits required for the assembly, stability, and functions of the BAF chromatin remodeling complex, we demonstrate that the BAF complex is essential for the proliferation, survival, and differentiation of the NCCs. Neural crest-specific deletion of *BAF155/BAF170* leads to embryonic lethality due to a wide range of developmental defects including craniofacial and cardiovascular defects. By performing RNAseq and transcription factor enrichment analysis we show that the BAF complex

Information files. RNA sequencing data have been deposited in GEO under accession code GSE153121.

**Funding:** This work was supported by funds from Duke-NUS Medical School Singapore and the Goh foundation and a Singapore National Research Foundation (NRF) fellowship (NRF-NRFF2016-01) to M.K.S. The funders had no role in study design, data collection and analysis, decision to publish, or preparation of the manuscript.

**Competing interests:** The authors have declared that no competing interests exist.

modulates the expression of multiple signaling pathway genes including Hippo and Notch, essential for the development of the NCCs. Furthermore, the BAF complex component physically interacts with the Hippo signaling components in NCCs to regulate gene expression. We demonstrated that the BAF complex is essential for the Brg1-Yap-Tead-dependent transcription of target genes in NCCs. Together, our results demonstrate a critical role of the BAF complex in modulating the gene regulatory network essential for the proper development of neural crest and neural crest-derived tissues.

## Introduction

Neural crest cells (NCCs) are a group of multipotent cells that are transiently generated along the vertebrate axis. In mammalian embryos, NCCs migrate from the dorsolateral edges of the neural plate before the formation of the neural tube. These NCCs undergo an epithelial-to-mesenchymal transition to delaminate and migrate through the extracellular space to multiple tissues and differentiate into various cell types including neurons, pigment cells, cartilage, bone, and smooth muscle of the cardiovascular system in the developing embryo. Depending upon their location along the anterior-posterior body axis and differentiation ability to form certain derivatives, NCCs can be subdivided into five axial populations: cranial, cardiac, vagal, trunk, and sacral NCCs. Cranial NCCs migrate and populate the face and the first and second pharyngeal arches, contributing to the cranial ganglia, craniofacial skeleton, palates, and other structures of the developing head. Cardiac NCCs migrate to the third, fourth, and sixth pharyngeal arches, differentiate into the smooth muscle cells, and contribute towards the remodeling of the pharyngeal arch arteries. Cardiac NCCs migrate further and participate in the septation of the cardiac OFT by contributing to the septum which divides the truncus arteriosus, into the aorta and pulmonary trunk. Defective cardiac NCCs development leads to various congenital cardiac malformations, such as disrupted pharyngeal arteries, pulmonary artery stenosis, persistent truncus arteriosus (due to failed septation of the OFT), and membranous ventricular septal defects [1,2]. Genetic or environmental factors affecting NCCs ability to proliferate, migrate, or differentiate leads to many congenital cardiovascular and craniofacial disorders and contribute to more than one-third of all congenital diseases in humans.

Epigenetic regulation at the chromatin level plays an important role during embryonic development by maintaining the size of the progenitor cell population, their survival, and differentiation into different cell lineages. Genetic mutations or alterations of epigenetic regulators and chromatin modulators cause congenital disorders. Over the years, the importance of epigenetic modulators in regulating neural crest development has been increasingly emphasized. However, the role of the Brg1/Brm-associated factors (BAF) complex, one of the ATP-dependent chromatin remodeling complexes in the neural crest is not well described and warrants further investigation. The structure of the BAF complex is composed of an ATPase, Brg1 or Brm, core structural subunits BAF155, BAF170, and other cell type-specific subunits [3,4]. Genetic deletion of BAF subunits such as *Brg1*, *BAF155*, or *BAF45* leads to early embryonic lethality [5–8]. During embryonic development, both BAF155 and BAF170 are ubiquitously expressed [9]. BAF155 is strongly expressed in proliferating stem/progenitor cells and weakly in differentiated cells [10–12]. In contrast, BAF170 is weakly expressed in proliferating stem/progenitor cells and strongly in differentiated cells [10–12]. Although low expression of BAF170 is detected in stem/progenitor cells and BAF155 in differentiated cells, this expression is necessary and sufficient for stabilizing the BAF complex [9,13]. During neural crest development, Brg1 promotes proliferation, migration, and differentiation of the NCCs into smooth

muscle cells as well as suppresses cell death. Brg1 physically interacts with another chromatin remodeler chromodomain-helicase-DNA-binding-protein 7 (Chd7) to promote expression of *PlexinA2*, a semaphorin receptor required for the proper migration of cardiac NCCs to the OFT [14]. Another study also demonstrated that Chd7 interaction with Polybromo-BAF (PBAF) in NCCs is essential for promoting neural crest-specific gene expression programs as well as for their migration. Other neural crest-related disease studies using mouse models have found that loss or mutation of BAF250a (*ARID1a*) leads to defects in the pharyngeal arteries, incomplete conotruncal septation of the OFT, and craniofacial malformation [15]. In contrast to the role of the BAF complex in other biological processes, its role in neural crest cells is not well described.

In the present study, using neural crest-specific deletion of *BAF155/170* we demonstrated the cell-autonomous role of the BAF complex in regulating NCCs migration, survival, proliferation, and differentiation. Previous reports have shown that the deletion of a single BAF subunit does not affect the expression and incorporation of other subunits into Brg1 or Brm-based complexes. To better understand the functions of the BAF complex, we deleted the core subunits of the BAF complex, BAF155, and BAF170 in the neural crest cells using $Wnt1^{Cre/+}$ and $Pax3^{Cre/+}$ mice and analyzed the neural crest-specific developmental defects. Loss of *BAF155/170* leads to embryonic lethality due to severe craniofacial and cardiovascular defects caused by impaired migration, survival, proliferation, and differentiation of NCCs. RNAseq analysis on sorted NCCs revealed that the BAF complex modulates the expression of many signaling pathway genes including Hippo and Notch pathways that are essential for the migration, proliferation, and differentiation of the NCCs. Transcription factor enrichment analysis identified several factors including Hey1, Hey2, and Hes5 that may directly or indirectly interact with the BAF complex in NCCs. Furthermore, we demonstrated that Brg1 physically interacts with the Yap and Tead transcription factors. Brg1-Tead and Yap-Tead interactions were significantly compromised in *BAF155/170* knockdown cells suggesting that the proper assembly of BAF complex is essential for Brg1-Yap-Tead-dependent transcription of target genes in NCCs. Together, these results reveal an essential role of the BAF complex in regulating neural crest-specific gene expression program required for the proper development of neural crest-derived tissues.

## Results

### Neural crest-specific deletion of *BAF155/170* results in craniofacial malformation and early embryonic lethality

BAF155 and BAF170 are expressed in the neural crest cells (S1 Fig). To determine the role of BAF complex in NCCs, we decided to genetically delete the core subunits BAF155 and BAF170 by crossing the $BAF155^{flox/flox}$ and $BAF170^{flox/flox}$ mice with two different neural crest Cre lines, $Pax3^{Cre/+}$ and $Wnt1^{Cre/+}$ mice [10,16,17]. Both $Pax3^{Cre/+}$ and $Wnt1^{Cre/+}$ alleles are active in NCCs. However, their cre expression domains are not restricted to the NCCs alone. For example, Wnt1 expression is observed in the dorsal neural stem cells that contribute to both the central nervous system and neural crest progenitors [18,19]. In contrast, $Pax3^{Cre/+}$ knock-in allele is active in the dorsal neural tube and somites of an early embryo and in the neural crest and somite derivatives of late gestation embryos [16,20]. Mice double heterozygous for *BAF155/170*-deficiency ($Pax3^{Cre/+};BAF155^{flox/+};BAF170^{flox/+}$ and $Wnt1^{Cre/+};BAF155^{flox/+};BAF170^{flox/+}$) appeared normal and were fertile. To characterize the phenotypic defects due to loss of *BAF155* and *BAF170*, embryos were harvested from E9.5 to E14.5. Analyses of embryos collected at different developmental stages revealed that $Pax3^{Cre/+}$-mediated deletion of both *BAF155* and *BAF170* ($Pax3^{Cre/+};BAF155^{flox/flox};BAF170^{flox/flox}$) leads to severe

craniofacial defects and embryonic death at E12.5 (Fig 1A–1L). *BAF155*-deficient (*Pax3$^{Cre/+}$; BAF155$^{flox/flox}$;BAF170$^{flox/+}$*) embryos were also affected but less severely than the *BAF155/ 170*-deficient (*Pax3$^{Cre/+}$;BAF155$^{flox/flox}$;BAF170$^{flox/flox}$*) embryos and die by E14.5. In contrast to the *BAF155/170*- or *BAF155*-deficient embryos, morphologically *BAF170*-deficient (*Pax3$^{Cre/+}$;BAF170$^{flox/flox}$;BAF155$^{flox/+}$*) embryos were largely indistinguishable from their littermate controls (Fig 1A, 1B, 1E, 1F, 1I and 1J). We also observed a hole in the forebrain suggesting an impaired development of telencephalon in *BAF155/170*- and *BAF155*-deficient embryos (Fig 1G, 1H, 1K and 1L). To determine the neural crest-specific requirement of the BAF complex, we generated neural crest-specific *BAF155/170*-deficient (*Wnt1$^{Cre/+}$;BAF155$^{flox/ flox}$;BAF170$^{flox/flox}$*) embryos using *Wnt1$^{Cre/+}$* mice (Fig 2A–2L) [20]. No significant changes were observed at E9.5 between different genotypes analysed (Fig 2A–2D) Consistent with the embryonic defects seen in *Pax3$^{Cre/+}$;BAF155$^{flox/flox}$;BAF170$^{flox/flox}$* embryos, *Wnt1$^{Cre/+}$; BAF155$^{flox/flox}$;BAF170$^{flox/flox}$* embryos also died at E12.5 and showed severe developmental defects including craniofacial deformities (Fig 2E–2L). We also observed hemorrhage in the forebrain of E11.5 *Wnt1$^{Cre/+}$;BAF155$^{flox/flox}$;BAF170$^{flox/flox}$* embryos (3/10 embryos), which were not observed in the *Pax3$^{Cre/+}$;BAF155$^{flox/flox}$;BAF170$^{flox/flox}$* embryos (Fig 2L). Compared to control or *BAF170*-deficient (*Wnt1$^{Cre/+}$;BAF170$^{flox/flox}$;BAF155$^{flox/+}$*) embryos, craniofacial defects were also evident in *BAF155*-deficient (*Wnt1$^{Cre/+}$;BAF155$^{flox/flox}$;BAF170$^{flox/+}$*) embryos (Fig 2E–2L). No obvious morphological differences were observed between control and *BAF170*-deficient (*Wnt1$^{Cre/+}$;BAF170$^{flox/flox}$;BAF155$^{flox/+}$*) embryos at E10.5 and E11.5 (Fig 2E, 2F, 2I and 2J).

## Impaired differentiation of NCCs into smooth muscle cells of the pharyngeal arch arteries due to *BAF155/170* deletion

To determine whether migration and differentiation of the NCCs are affected due to *BAF155* and *BAF170* deletion, we performed the lineage-tracing analysis at E9.5, E10.5 and E11.5 in both controls, *BAF155*-deficient (*Wnt1$^{Cre/+}$;BAF155$^{flox/flox}$;BAF170$^{flox/+}$*), *BAF170*-deficient (*Wnt1$^{Cre/+}$;BAF170$^{flox/flox}$;BAF155$^{flox/+}$*) and *BAF155/170*-deficient (*Wnt1$^{Cre/+}$;BAF155$^{flox/flox}$; BAF170$^{flox/flox}$*) embryos (Fig 3A–3L). Labeled neural crest cells marked by GFP expression were abundantly present around the neural tube and in the craniofacial tissues in *BAF155*- and *BAF170*-deficient as well as in *BAF155/170*-deficient embryos (Fig 3B–3D, 3F–3H and 3J– 3L). GFP$^+$ NCCs were also seen in the developing pharyngeal arches of *BAF155/170*-deficient embryos. Whole embryo immunofluorescence analysis did not reveal any gross differences regarding the migration of NCCs. To get a better understanding of NCCs migration, we also generated frontal sections of lineage-traced E11.5 embryos of all the genotypes and performed GFP immunostaining (Fig 3M–3P). Consistent with the earlier observation, immunostaining with anti-GFP antibody also revealed no difference in the migration of the NCCs at the level of pharyngeal arches in *BAF155*-, *BAF170*- and *BAF155/170*-deficient embryos (Fig 3M–3P). Since NCCs in the pharyngeal arches contribute to the vascular smooth muscle cells of the pharyngeal arch arteries, we decided to determine whether the disruption of the BAF complex affects the ability of NCCs to differentiate into smooth muscle cells. We performed immunostaining on the frontal sections of E11.5 embryos for GFP, to mark the neural crest derivatives and SM22α, a smooth muscle cell marker. Dapi staining was performed to visualize the nuclei (Fig 3Q–3T). In control embryos, SM22α$^+$ smooth muscle cells covered most of the vascular walls suggesting proper differentiation of NCCs (Fig 3Q). Compared to the controls, no significant change was observed in *BAF170*-deficient embryos (Fig 3R). In *BAF155*-deficient embryos, the pharyngeal arches were poorly developed and SM22α$^+$ cells were observed but not properly distributed throughout the vascular wall (Fig 3S). In contrast to the *BAF155*- or

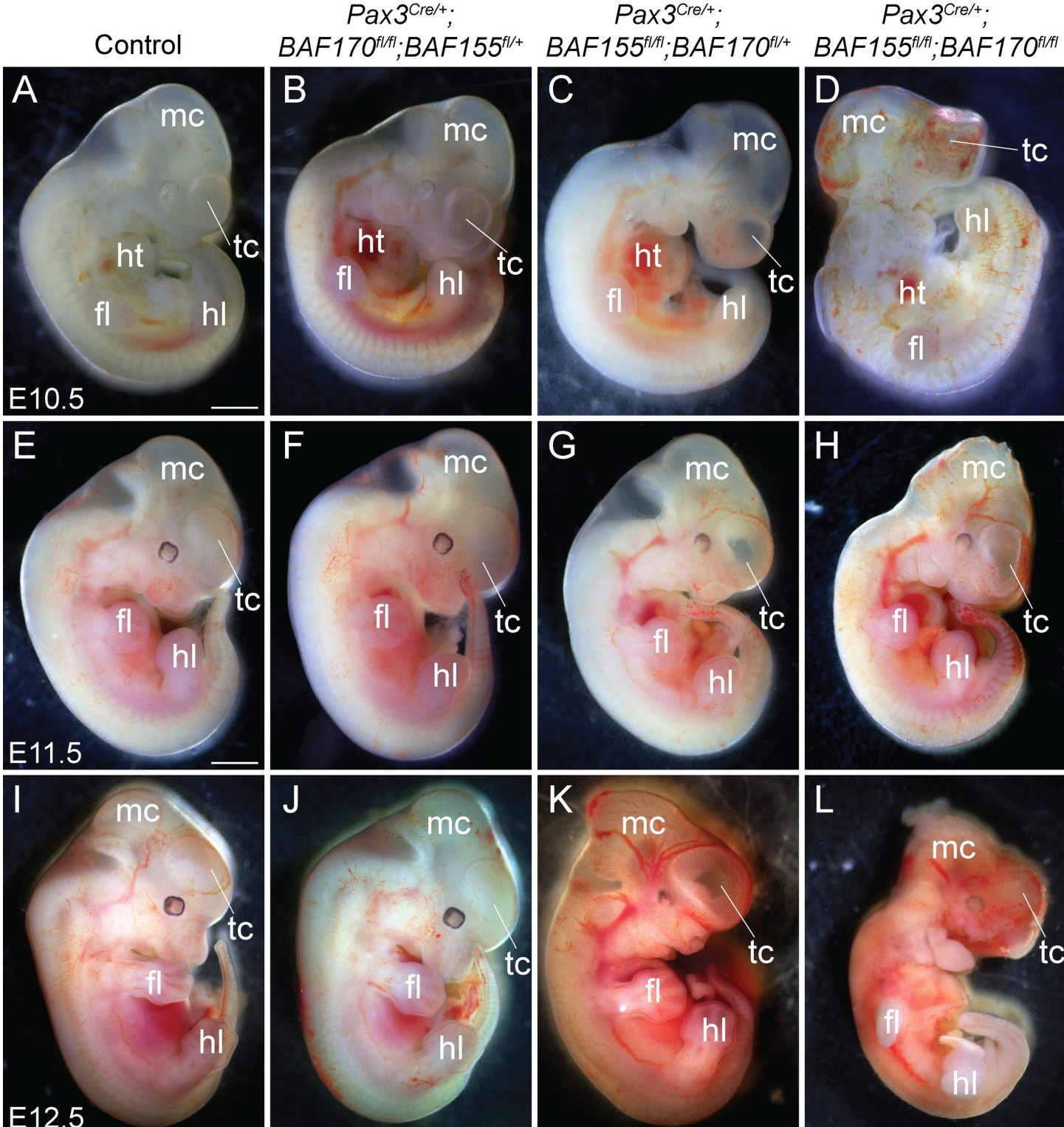

**Fig 1. *Pax3^Cre*-mediated deletion of *BAF155/170* leads to severe craniofacial defects and embryonic lethality. (A-L)** Phenotypic defects in control, *BAF170*-deficient (*Pax3^Cre/+^;BAF170^fl/fl^;BAF155^fl/+^*), *BAF155*-deficient (*Pax3^Cre/+^;BAF155^fl/fl^;BAF170^fl/+^*) and *BAF155/170*-deficient (*Pax3^Cre/+^;BAF155^fl/fl^;BAF170^fl/fl^*) embryos. Sagittal view of E10.5 (A-D), E11.5 (E-H), and E12.5 (I-L) embryos. Severe craniofacial defects in *BAF155/170*-deficient embryos (D, H, and L) when compared with their littermate controls (A, E, and I). Among *BAF155/170*-deficient embryos, 9/9 showed forebrain defects, evident by improper development of the telencephalon (D, H, and L). *BAF155*-deficient embryos (C, G, and K) also demonstrated abnormal development of craniofacial tissues and telencephalon when compared with their littermate controls

(A, E, and I). No obvious morphological defects were observed in *BAF170*-deficient embryos at the embryonic stages analyzed (B, F, and J). n = 3–10 embryos were analyzed for each genotype at each given embryonic stage. Scale bars 200μM (A-D) and 500μM (E-L) fl, forelimb; hl, hindlimb; ht, heart; mc, metencephalon; tc, telencephalon.

*BAF170*-deficient embryos, SM22α$^+$ cells were nearly absent from the vascular walls of *BAF155/170*-deficient embryos suggesting impaired differentiation of the NCCs into smooth muscle cells (Fig 3T). Consistent with the smooth muscle defects seen in *Wnt1$^{Cre/+}$;BAF155$^{flox/flox}$;BAF170$^{flox/flox}$* embryos, *Pax3$^{Cre/+}$;BAF155$^{flox/flox}$;BAF170$^{flox/flox}$* embryos also showed a severe deficiency of SM22α$^+$ cells in the vascular wall of the pharyngeal arch arteries (S2 Fig). The decreased formation of smooth muscle wall around the pharyngeal arteries suggests that the BAF complex is essential for the differentiation of NCCs into smooth muscle cells.

## Impaired contribution of cardiac NCCs in the OFT due to *BAF155/170* deletion

After passing through the pharyngeal arches, the cardiac neural crest cells continue to migrate toward the heart and eventually contribute to the formation of the aortopulmonary septum as well as the smooth muscle layer of the OFT [21]. To determine whether NCCs contribution to the developing OFT was affected due to *BAF155/170* deletion, we performed the lineage-tracing experiment and analyzed the cardiac NCCs contribution by immunostaining the sagittal sections of E11.5 control and *BAF155/170*-deficient embryos through the OFT with anti-GFP antibody and counter-staining the nuclei with Dapi (Fig 4A–4F). In control embryos, the GFP$^+$ population of NCCs was readily apparent in the conotruncal cushions of the developing OFT (Fig 4A and 4B). In contrast to the normal migration of NCCs in the pharyngeal arches, *BAF155/170*-deficient cardiac NCCs failed to migrate properly to the heart, as evidenced by the quantitative analysis showing reduced distance migrated by the NCCs as well as a reduced number of GFP$^+$ cells in the conotruncal cushions of the developing OFT (Fig 4A–4F). As an additional method to mark these migrating NCCs cells to the OFT, we performed immunostaining for PlexinA2, a marker of migratory and post-migratory cardiac NCCs, and Dapi counter-staining to visualize the nuclei (Fig 4G and 4H). When compared with controls, *BAF155/170*-deficient embryos showed reduced levels of PlexinA2 expression in the conotruncal cushions (Fig 4H). To demonstrate the importance of *BAF155/170* in smooth muscle cell recruitment and/or differentiation in the OFT, we performed immunostaining for SM22α, a smooth muscle cell marker, and Dapi counter-staining to visualize the nuclei (Fig 4I and 4J). As expected, our results showed SM22α$^+$ cells surrounding the OFT and populating the conotruncal cushions in the control embryos (Fig 4I). In *BAF155/170*-deficient embryos, we observed SM22α$^+$ cells surrounding the OFT. However, the number of SM22α$^+$ cells present in the conotruncal cushions was considerably decreased compared to the controls (Fig 4J). This decrease in the number of SM22α$^+$ cells in the conotruncal cushions was due to a reduced number of migrating GFP$^+$ NCCs and likely not due to impaired differentiation (Fig 4K and 4L). Together these findings suggest that BAF complex function is essential for the proper contribution of the cardiac NCCs to the developing OFT.

As the deletion of *BAF155* alone or in combination with *BAF170* using both Cre lines resulted in early embryonic lethality, we could not analyze other neural crest-derived tissue such as secondary palates in these embryos. During development, signaling between palate epithelium and underlying neural crest-derived mesenchyme is essential for the growth of palate shelves and the formation of the secondary palate [22–25]. To determine the role of the BAF complex in secondary palate formation, we analyzed the *BAF170*-deficient (*Wnt1$^{Cre/+}$;BAF170$^{flox/flox}$;BAF155$^{flox/+}$*) at E15.5 and observed cleft palate (S3 Fig). We also utilized a

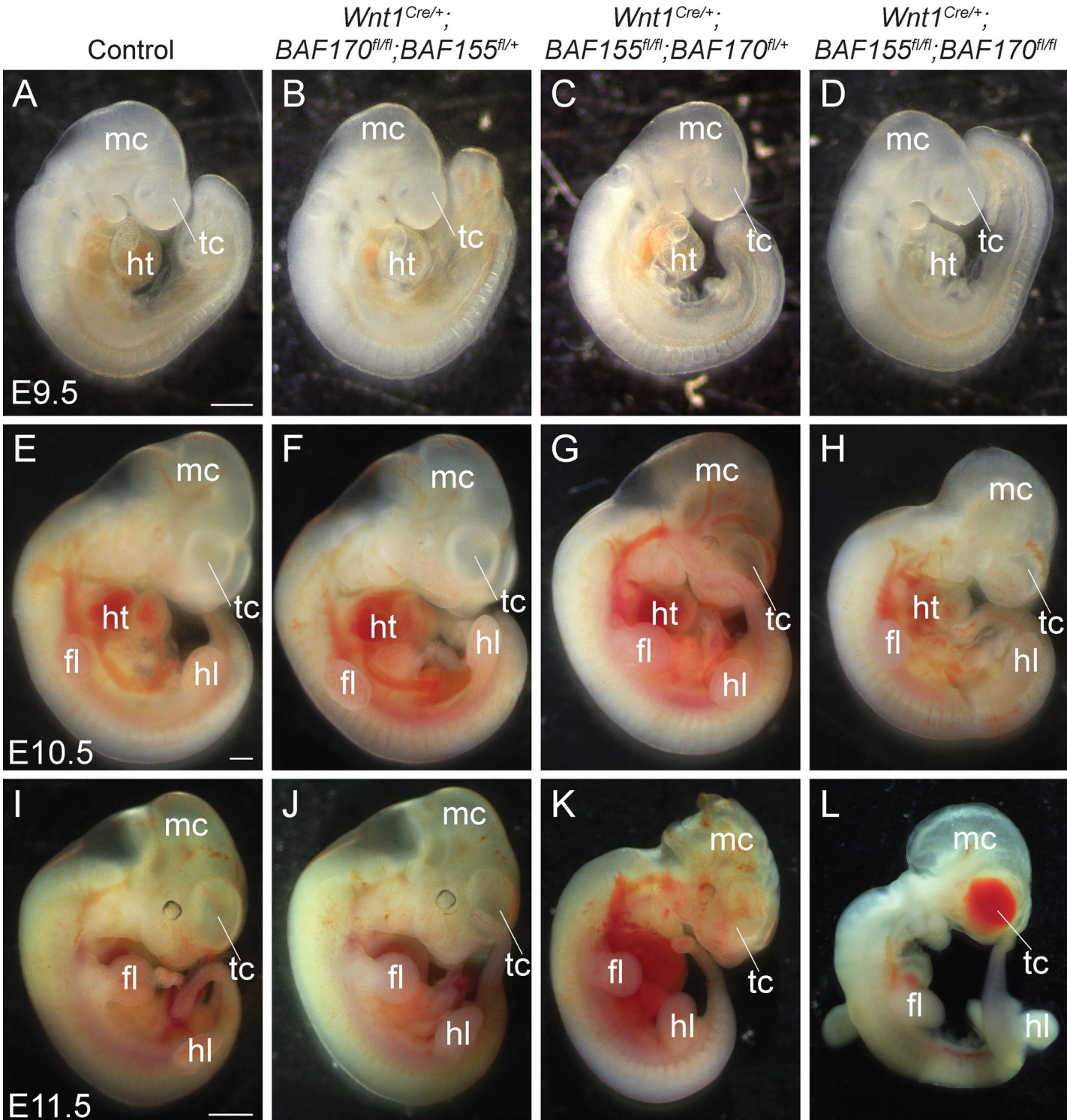

**Fig 2. *Wnt1^Cre*-mediated neural crest-specific deletion of *BAF155/170* leads to severe craniofacial defects and embryonic lethality. (A-H)** Phenotypic defects in control, *BAF170*-deficient (*Wnt1^Cre/+^;BAF170^fl/fl^;BAF155^fl/+^*), *BAF155*-deficient (*Wnt1^Cre/+^;BAF155^fl/fl^;BAF170^fl/+^*) and *BAF155/170*-deficient (*Wnt1^Cre/+^;BAF155^fl/fl^;BAF170^fl/fl^*) embryos. Sagittal view of E9.5 (A-D), E10.5 (E-H), and E11.5 (I-L) embryos. No obvious developmental defects were observed at E9.5 between different genotypes (A-D). Developmental defects in neural-crest derived tissues, including craniofacial defects in *BAF155/170*-deficient embryos (H and L) when compared with their littermate controls (E and I). Among *BAF155/170*-deficient embryos, 7/10 showed severe craniofacial defects. Hemorrhage in the forebrain (telencephalon) of E11.5 *BAF155/170*-deficient embryos (3/10 embryos) (L). Craniofacial defects are also present in *BAF155*-deficient embryos (6/10 showed craniofacial deformities, enlarged blood vessels) but less severe than the *BAF155/170*-deficient embryos (K). No obvious morphological defects were observed in *BAF170*-deficient embryos (10/

10) at the embryonic stages analyzed (F and J). n = 4–10 embryos were analyzed for each genotype at each given embryonic stage. Scale bars 200μM (A-H) and 500μM (E-H). fl, forelimb; hl, hindlimb; ht, heart; mc, metencephalon; tc, telencephalon.

recently reported $BAF155^{FoxG1-CKO}$ ($FoxG1^{Cre/+}$;$BAF155^{fl/fl}$) mutant mice and observed cleft palate in E15.5 embryos (S4 Fig) [17,26,27]. In the $FoxG1^{Cre/+}$ line, Cre recombinase activity is driven in discrete structures of the head including palate epithelium [28]. These findings suggest that the BAF complex also plays an important role in the neural crest-derived palate mesenchyme as well as in the palate epithelium during the secondary palate formation. In addition to these defects, other neural crest derivatives were also affected due to $BAF155/170$ deletion. Whole-mount neurofilament (2H3) staining showed impaired NCCs differentiation into neurons (S5 Fig).

## Decreased NCCs cell proliferation and increased apoptosis due to loss of $BAF155/170$

To determine the cellular consequences due to $BAF155/170$ deletion, we performed proliferation and apoptotic assay on E9.5 and E10.5 control and $BAF155/170$-deficient ($Wnt1^{Cre/+}$; $BAF155^{flox/flox}$;$BAF170^{flox/flox}$) embryo sections. We performed immunostaining for Ki67 and quantified the percentage of proliferating cells (ratio of Ki67-positive cells to the total number of cells as determined by Dapi counterstaining in the defined area) in the neural tube and pharyngeal arches (Fig 5A–5E). No significant change in NCCs proliferation was observed at E9.5 (Fig 5E). However, at E10.5, NCCs proliferation was significantly reduced in the $BAF155/170$-deficient embryos compared to the controls in both the areas quantified (Fig 5C–5E). To determine the changes in cell apoptosis due to BAF deletion, we performed terminal deoxynucleotidyl transferase dUTP nick end labeling (TUNEL) analysis on sections from E9.5 and E10.5 control and $BAF155/170$-deficient embryos and quantified the percentage of TUNEL positive cells in the neural tube and pharyngeal arch areas (Fig 5F–5J). We observed a significant increase in cell death around the neural tube as well as in the pharyngeal arches of $BAF155/170$-deficient embryos at both time points analyzed (Fig 5H–5J). Consistent with changes in the number of apoptotic cells in $Wnt1^{Cre/+}$;$BAF155^{flox/flox}$;$BAF170^{flox/flox}$ embryos, we also observed increased apoptosis in $Pax3^{Cre/+}$;$BAF155^{flox/flox}$;$BAF170^{flox/flox}$ embryos (Fig 5K–5O). We also observed increased apoptosis in $Pax3^{Cre/+}$-derived non-neural crest-derived tissues such as somites, due to $BAF155/170$ deletion. Similar to neural tube and pharyngeal arches, reduced proliferation and increased apoptosis was also observed in $BAF170$-deficient ($Wnt1^{Cre/+}$;$BAF170^{flox/flox}$;$BAF155^{flox/+}$) palate shelves, likely causing cleft palate (S3 Fig). These findings suggest that BAF complex function is essential for NCCs proliferation and survival.

## BAF complex is critical for controlling neural crest-specific gene expression

To determine the molecular changes associated with the loss of $BAF155/170$ in NCCs, RNAseq analysis was performed on GFP$^+$ NCCs isolated from E11.25 control ($Wnt1^{Cre/+}$;$BAF155^{flox/+}$; $BAF170^{flox/+}$;$Rosa26^{mTmG/+}$) and $BAF155/170$-deficient ($Wnt1^{Cre/+}$;$BAF155^{flox/flox}$;$BAF170^{flox/flox}$;$Rosa26^{mTmG/+}$) embryos using fluorescence-activated cell sorting (FACS) technique (Figs 6A and S6). Paired-end RNAseq reads from each sample (median read depth > 32 million) were adapter-trimmed (to <10% adapter sequences in a read) and aligned to the mouse reference genome. A total of 3011 genes were differentially expressed between the control and $BAF155/170$-deficient groups, with 2046 down-regulated and 965 up-regulated genes (Fig 6B and S1 Table). RNAseq analysis showed that genes down-regulated in $BAF155/170$-deficient

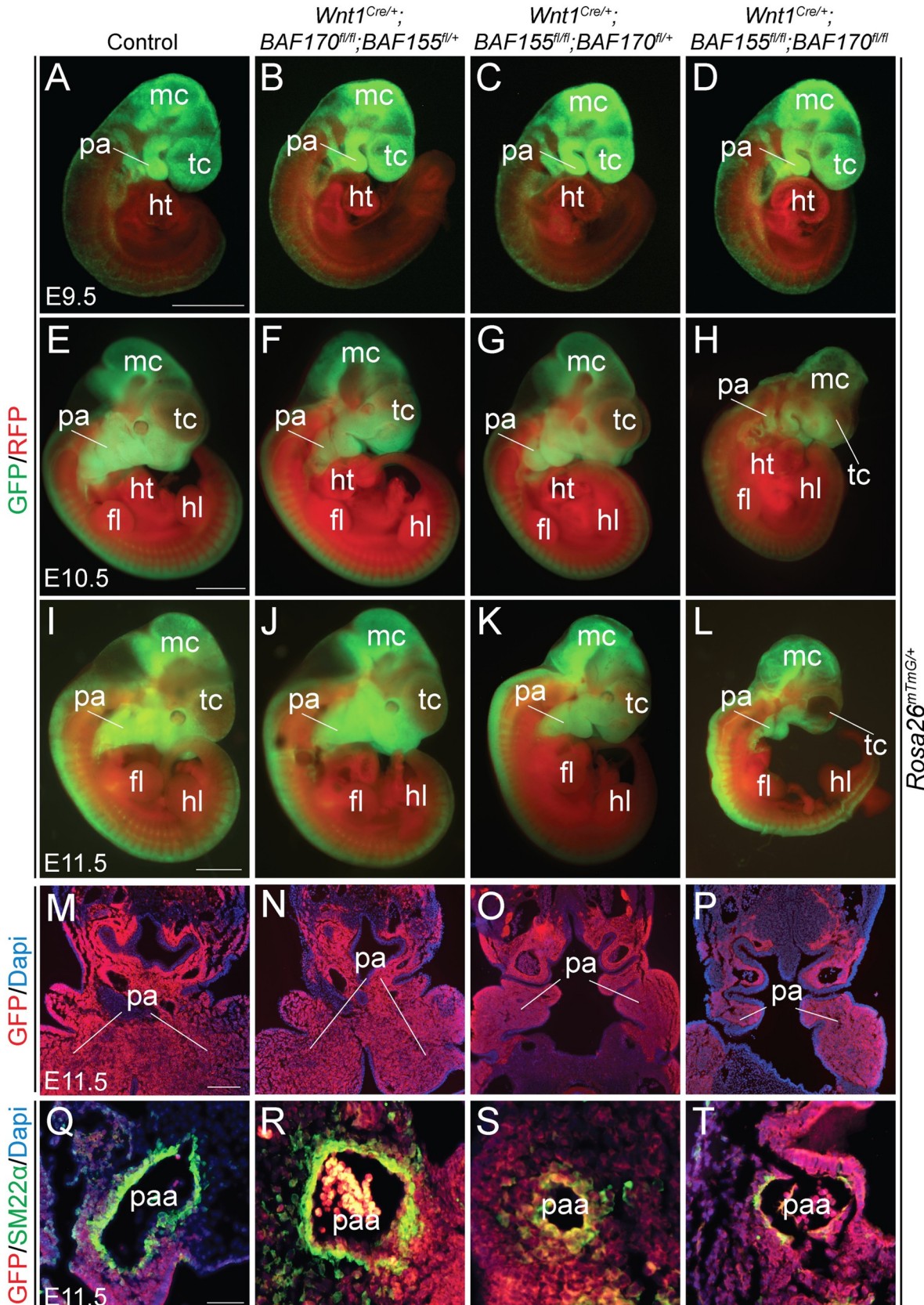

**Fig 3. Defective NCCs differentiation in the pharyngeal arch arteries of neural crest-specific *BAF155/170* knockout embryos. (A-H)** Lineage tracing of *Wnt1^{Cre/+}*-derived NCCs using *R26^{mTmG/+}* reporter at E9.5 (A-D), E10.5 (E-H) and E11.5 (I-L). Control, *BAF170*-deficient (*Wnt1^{Cre/+};BAF170^{fl/fl};BAF155^{fl/+}*), *BAF155*-deficient (*Wnt1^{Cre/+};BAF155^{fl/fl};BAF170^{fl/+}*) and *BAF155/170*-deficient (*Wnt1^{Cre/+}; BAF155^{fl/fl};BAF170^{fl/fl}*) embryos were analyzed by GFP and RFP immunofluorescence (n = 4 each genotype). Merged GFP/RFP images are presented here (A-L). Scale bars 200μM (A-H) and 500μM (I-L). **(M-P)** Anti-GFP immunostaining on the frontal sections of E11.5 control (M), *BAF170*-deficient (N), *BAF155*-deficient (O), and *BAF155/170*-deficient (P) embryos showing neural crest-derived cells in the pharyngeal arches (n = 3 each genotype). Scale bar 75μM (M-P). **(Q-T)** Anti-SM22α immunostaining on the frontal sections of E11.5 control (Q), *BAF170*-deficient (R), *BAF155*-deficient (S), and *BAF155/170*-deficient (T) embryos showing differentiation of neural crest-derived cells into the smooth muscle of the 4^{th} pharyngeal arch artery (n = 3 each genotype). Scale bar 50μM (Q-T). fl, forelimb; hl, hindlimb; ht, heart; mc, metencephalon; pa, pharyngeal arch; paa, pharyngeal arch artery; tc, telencephalon.

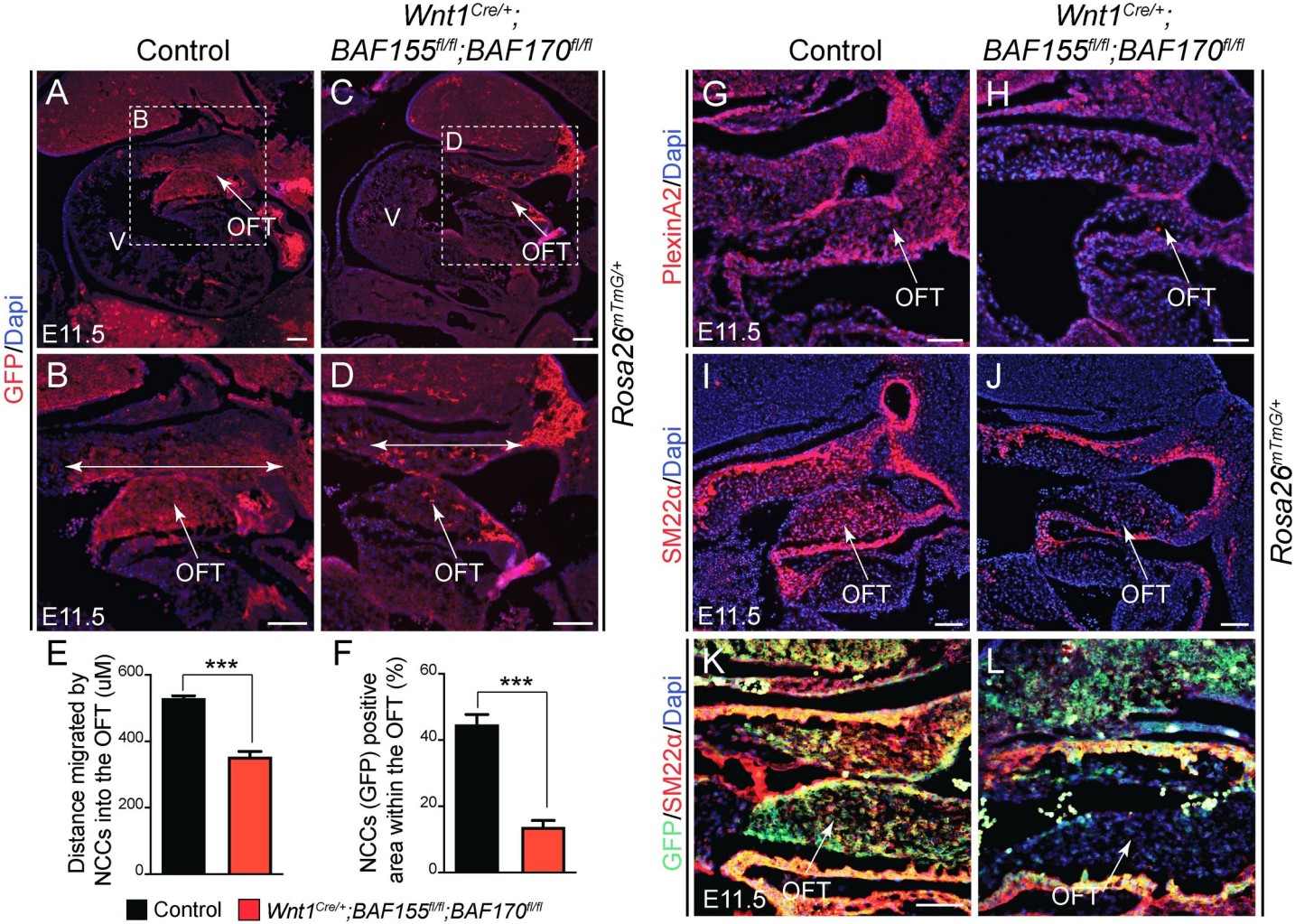

**Fig 4. Impaired NCCs contribution to the developing cardiac outflow tract (OFT) of the neural crest-specific *BAF155/170* mutant embryos. (A-D)** Lineage tracing of *Wnt1^{Cre/+}*-derived cardiac NCCs using *R26^{mTmG/+}* reporter in E11.5 control and *BAF155/170*-deficient (*Wnt1^{Cre/+};BAF155^{fl/fl};BAF170^{fl/fl}*) embryos. Anti-GFP immunostaining on the sagittal sections showing recruited cardiac NCCs in the OFT of control (A-B) and *BAF155/170*-deficient (C-D) embryos. Nuclei were visualized by Dapi staining (blue). Double head arrows show the distance of migrated cardiac NCCs in the OFT (B and D). **(E)** Quantification of distance migrated by NCCs into the OFT (n = 3). **(F)** Quantification of GFP+ neural crest cell area in the OFT (n = 3). **(G-H)** anti-PlexinA2 immunostaining and Dapi counterstaining on the OFT sections of control (E) and *BAF155/170*-deficient (F) embryos. **(I-J)** anti-SM22α immunostaining and Dapi counterstaining on OFT sections of control (G) and *BAF155/170*-deficient (H) embryos. (K-L) Double immunostaining for GFP and SM22α on OFT sections of control (K) and *BAF155/170*-deficient (L) embryos. (n = 3–4 each genotype). Values are reported as means ± SEM (*P < 0.05, **P < 0.01, ***P < 0.001; NS, not significant). Scale bar 75μM (A-L). OFT, outflow tract; V, ventricle.

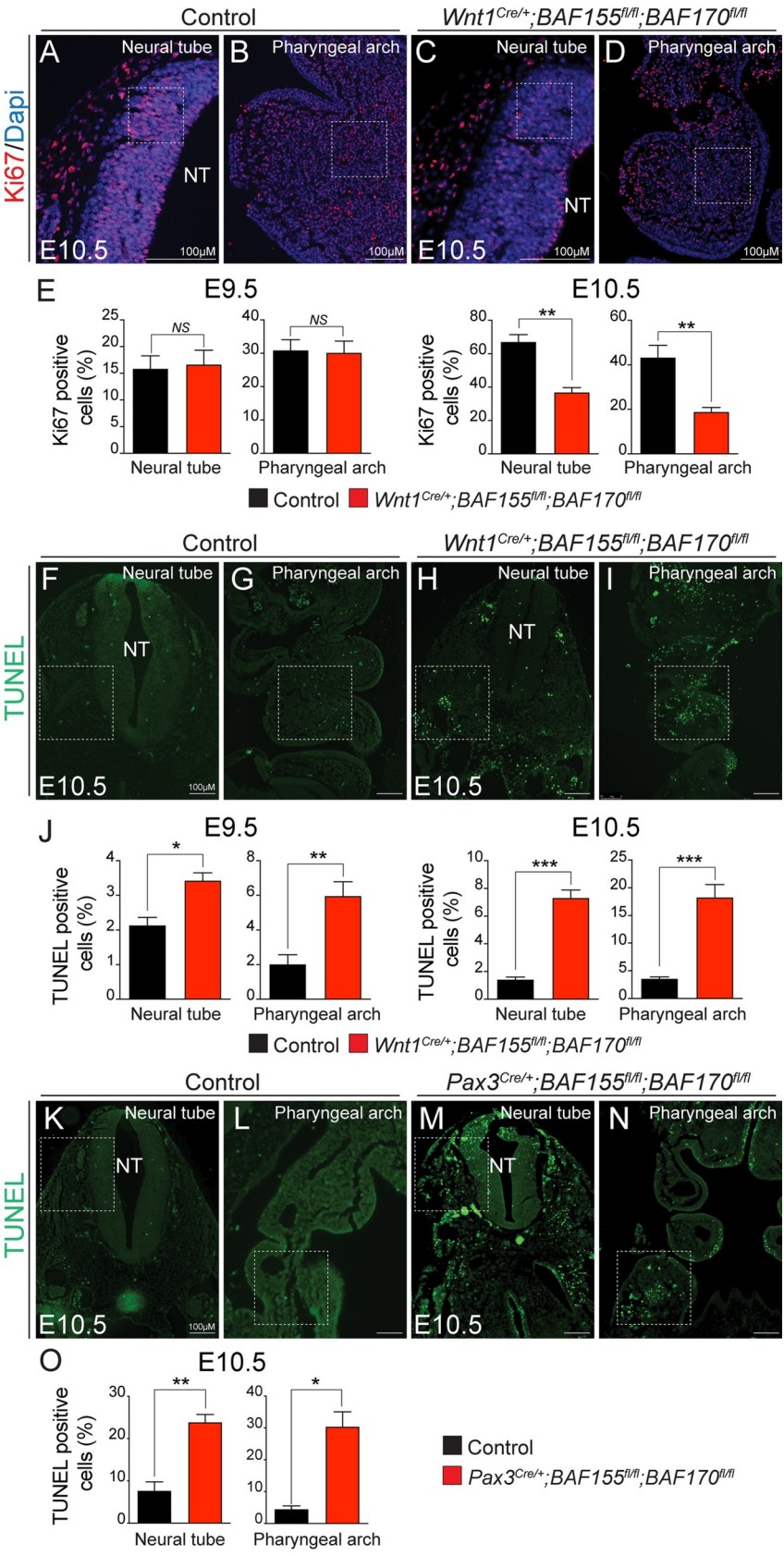

**Fig 5. Increased apoptosis and decreased cell proliferation in *BAF155/170^Wnt1-CKO* embryos. (A-D)** Anti-Ki67 immunostaining and Dapi counterstaining on frontal sections with neural tube and pharyngeal arch area of E9.5 and E10.5 control and *BAF155/170*-deficient (*Wnt1^Cre/+;BAF155^fl/fl;BAF170^fl/fl*) embryos. **(E)** Quantification of cell proliferation was calculated as the ratio of Ki67-positive cells to the total number of cells as determined by Dapi counterstaining in the defined area of the neural tube and pharyngeal arch (E). (n = 3–4 each genotype). **(F-J)** TUNEL assay and quantification was performed on E9.5 and E10.5 control and *BAF155/170*-deficient (*Wnt1^Cre/+;BAF155^fl/fl; BAF170^fl/fl*) sections. (n = 3–4 each genotype). **(K-O)** TUNEL assay and quantification was performed on E10.5 control and *BAF155/170*-deficient (*Pax3^Cre/+;BAF155^fl/fl;BAF170^fl/fl*) sections. (n = 3–4 each genotype). Values are reported as means ± SEM (*P < 0.05, **P < 0.01, ***P < 0.001; NS, not significant).

NCCs were substantially enriched in craniofacial development, OFT development, NCCs proliferation, differentiation and migration, neural crest-specification and axon guidance; whereas genes up-regulated in *BAF155/170*-deficient NCCs were enriched in the inflammatory response and positive regulators of apoptotic pathways (Fig 6B). Consistent with the reduced proliferation and increased apoptosis observed in *BAF155/170* mutant embryos, gene ontology analysis using Enrichr showed that genes involved in cell cycle regulation were downregulated, and genes involved in apoptosis were upregulated (Fig 7A). MA plot and Enrichr analysis also demonstrated that genes required for neural crest cell proliferation, migration, and differentiation were significantly altered due to *BAF155/170* deletion (Figs 7B and S7). For example, Notch and Hippo, the two most important signaling pathways required for NCCs proliferation, migration, and differentiation were downregulated in *BAF155/170*-deficient NCCs (Fig 7B–7D). These changes were further validated by quantitative RT-PCR using RNA isolated from sorted NCCs. We observed reduced expression of Notch pathway genes such as *Jag2*, *Notch1*, *Notch3*, *Hes1*, *Hes5*, and *HeyL* (Fig 7B and 7C). Expression of genes that are part of the Hippo signaling pathway such as *Yap*, *Tead1*, *Tead2*, *Tead3*, *Amot*, and *Axin2* was also reduced due to *BAF155/170* deletion (Figs 7B–7D and S8). We also observed a reduction in the *PlexinA2* expression, a known Brg1 target in NCCs (Fig 7E). To determine the positive regulation of the Notch pathway by BAF complex, we performed luciferase activation assay using *Hes1*-luciferase construct in O9-1 cells with control siRNA or *BAF155/170* siRNA (S9 Fig). We observed increased activity of the *Hes1*-luciferase construct when transfected with NICD expression plasmids. Compared with O9-1 cells transfected with control siRNA, *BAF155/170* knockdown O9-1 cells had significantly reduced levels of *Hes1*-luciferase activity in response to NICD (S9 Fig).

## BAF complex is essential for Brg1-Yap-Tead-dependent transcription of target genes in NCCs

To determine the interaction between the BAF complex and Hippo signaling, we transfected O9-1 cells with control siRNA, *BAF155* siRNA, or *BAF155/170* siRNA, followed by immunoprecipitation using pan-Tead antibody and western blot for Brg1 (Fig 8A and 8B). The knockdown efficiency of *BAF155* and *BAF170* in O9-1 cells was confirmed by western blot analysis (Fig 8A). The co-immunoprecipitation experiment revealed that Brg1 interacts with Tead factors in O9-1 cells and this interaction is disrupted in *BAF155* knockdown or *BAF155/170* double knockdown conditions (Fig 8B). To determine the interaction between Brg1 and Yap, we performed immunoprecipitation using Brg1 antibody and western blot for Yap (Fig 8B). We observed that Brg1 interaction with Yap is relatively unaffected in *BAF155* knockdown or *BAF155/170* double knockdown cells (Fig 8B). Yap interacts with Tead transcription factors to regulate target gene expression. To determine the impact of *BAF155* or *BAF155/170* knockdown on Yap-Tead interaction, we performed immunoprecipitation using pan-Tead antibody followed by western blot for Yap. The co-immunoprecipitation experiment revealed that Yap interaction with Tead is severely disrupted in *BAF155* knockdown or *BAF155/170* double knockdown conditions (Fig 8B). To determine the positive regulation of the Hippo pathway

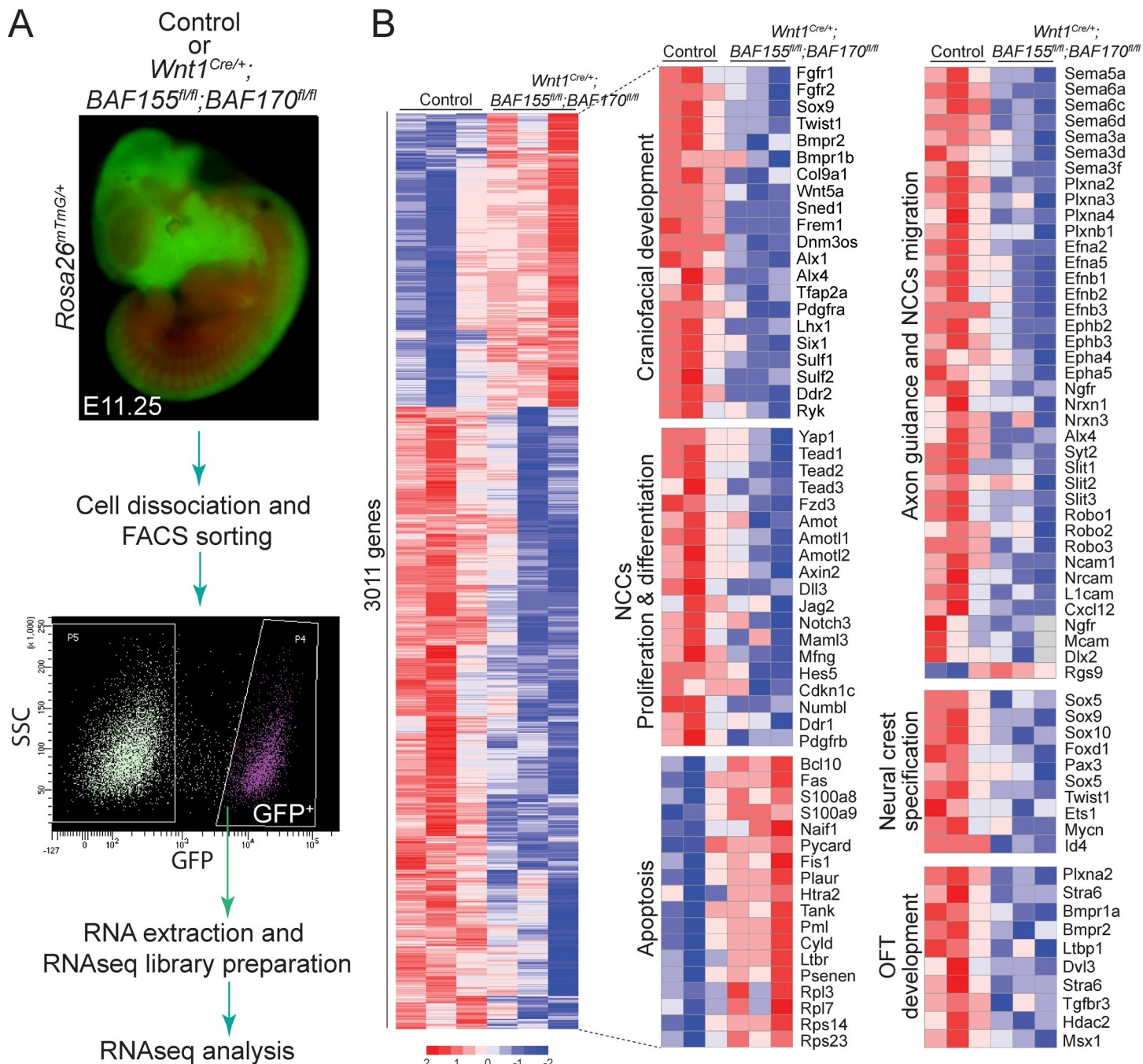

**Fig 6. Gene expression changes in isolated NCCs due to *BAF155/170* deletion.** (A) Experimental design for RNAseq analysis. Single-cell suspensions were prepared from lineage traced control (*Wnt1^Cre/+^:BAF155^flox/+^:BAF170^flox/+^:Rosa26^mTmG/+^*) and *BAF155/170*-deficient (*Wnt1^Cre/+^:BAF155^flox/flox^:BAF170^flox/flox^:Rosa26^mTmG/+^*) embryos for Fluorescence-activated cell sorting (FACS). NCCs that were positive for GFP were collected from control and *BAF155/170*-deficient embryos for RNA isolation and subsequently for library preparation and RNAseq analysis. (B) Heat map of differentially expressed transcripts (3011 genes) from RNAseq analysis of sorted control and *BAF155/170*-deficient NCCs. Genes regulating different aspects of neural crest biology including proliferation and differentiation were significantly down-regulated in *BAF155/170*-deficient NCCs compared to the controls. In contrast, genes regulating apoptosis were up-regulated in *BAF155/170*-deficient NCCs compared to controls. (n = 3 each genotype).

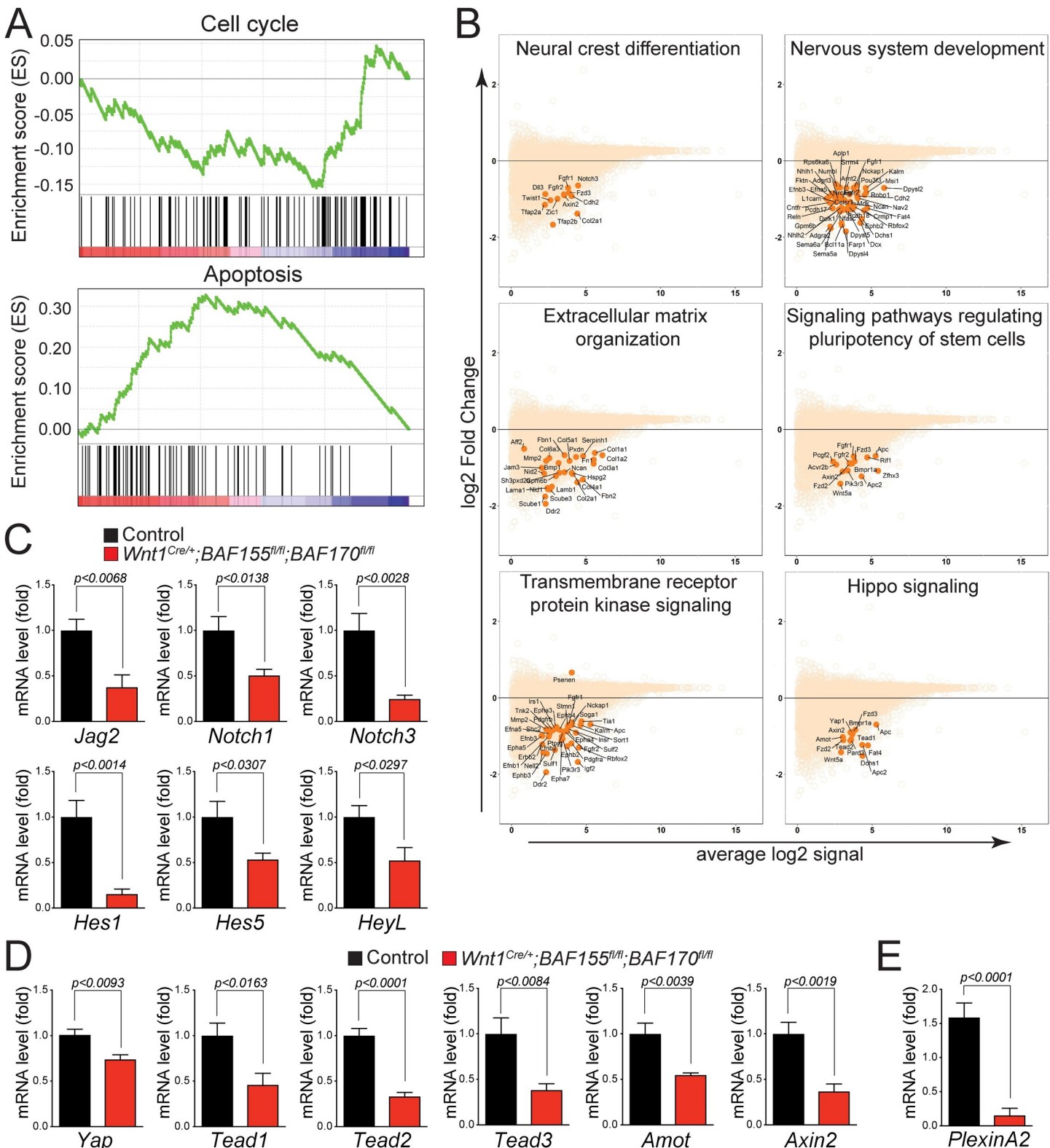

**Fig 7. Signaling pathways regulating NCCs proliferation and differentiation are downregulated in *BAF155/170* mutants. (A)** GSEA analysis showed that genes controlling cell cycle progression were negatively enriched while genes controlling apoptosis were positively enriched. **(B)** MA plots showing down-regulation of genes associated with critical pathways essential for NCCs proliferation, migration, and differentiation. **(C-D)** Relative mRNA levels of Notch (C) and Hippo (D) pathway genes in control and *BAF155/170*-deficient (*Wnt1^{Cre/+};BAF155^{fl/fl};BAF170^{fl/fl}*) sorted NCCs. n = 3–4 each genotype. **(E)** Relative mRNA levels of *PlexinA2* in control and *BAF155/170*-deficient (*Wnt1^{Cre/+};BAF155^{fl/fl};BAF170^{fl/fl}*) sorted NCCs. Values are reported as means ± SEM (*P < 0.05, **P < 0.01, ***P < 0.001; NS, not significant).

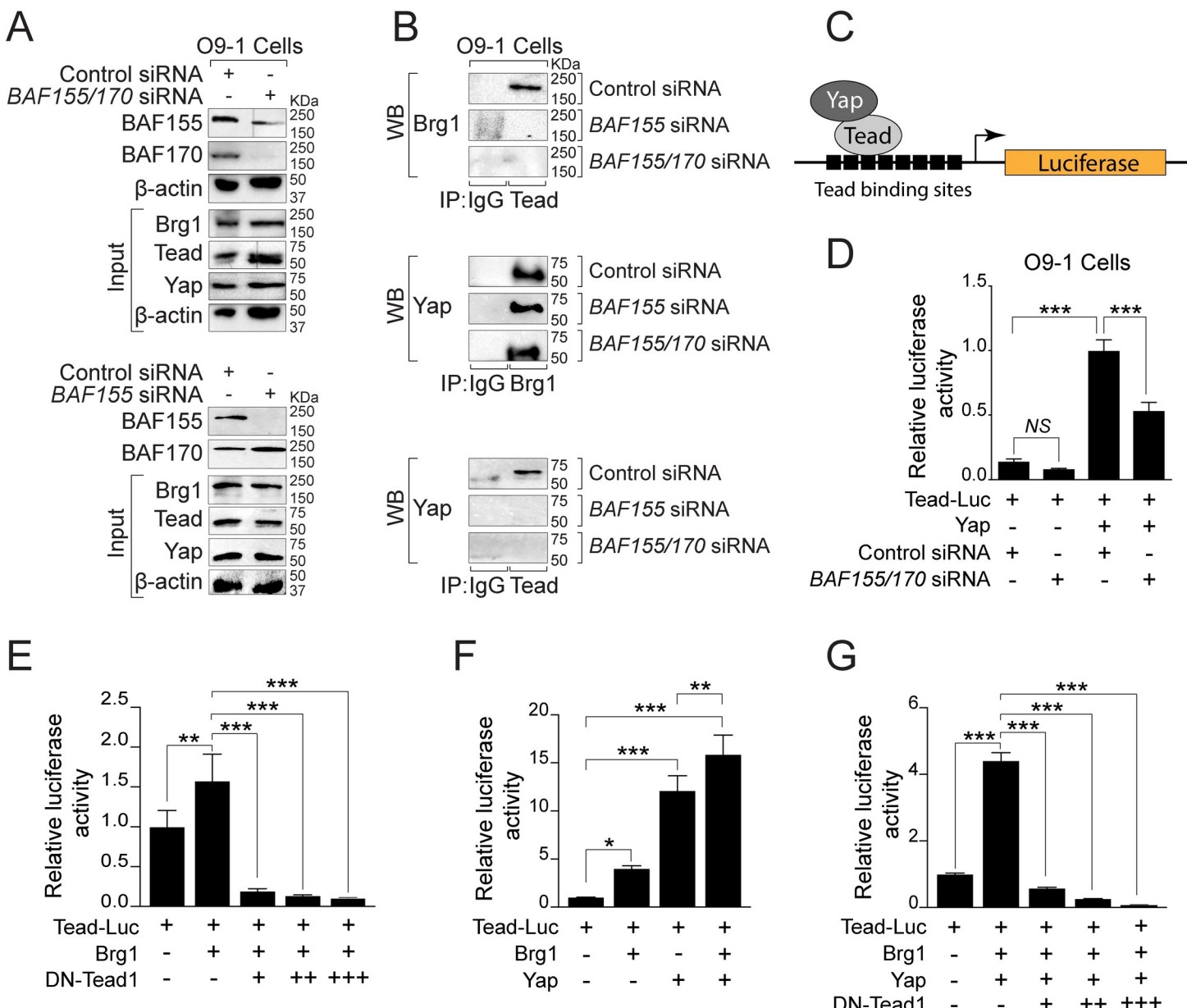

**Fig 8. BAF complex is essential for Brg1-Yap-Tead-dependent transcription of target genes in NCCs. (A)** Western blot for BAF155 and BAF170 using protein extracts from O9-1 cells transfected with control, *BAF155* or *BAF155/170* siRNA for 72 hours. Knockdown efficiency was evaluated by BAF155 and BAF170 western blots. Input blots for Brg1, Tead, and Yap together with β-actin controls are presented. **(B)** Cell extracts from control siRNA, *BAF155* siRNA, and *BAF155/170* siRNA transfected cells were prepared for immunoprecipitation (IP) using IgG control and anti-pan Tead antibody followed by western blotting (WB) for Brg1. IP was performed using IgG control and anti-Brg1 antibody followed by WB for Yap. Similarly, IP was performed using IgG control and anti-pan Tead antibody followed by WB for Yap. Representative blots are presented. **(C)** Schematic illustration of the Tead luciferase reporter construct with Tead and Yap. **(D)** Results of normalized luciferase reporter assays in O9-1 cells with Tead-luciferase reporter (Tead-Luc) in the presence of control siRNA or *BAF155/170* siRNA. **(E)** Results of normalized luciferase reporter assays in HEK293T cells with Tead-Luc reporter in the presence of Brg1 alone or in combination with dominant-negative Tead1 (DN-Tead1). **(F)** Results of normalized luciferase reporter assays in HEK293T cells with Tead-Luc reporter in the presence of Brg1, Yap alone, or both. **(G)** Results of normalized luciferase reporter assays in HEK293T cells with Tead-Luc reporter in the presence of Brg1 and Yap in combination with different doses of DN-Tead1. Values are reported as means ± SD (*$P < 0.05$, **$P < 0.01$, ***$P < 0.001$; NS, not significant).

by the BAF complex, we performed luciferase activation assay using *Tead*-luciferase construct in O9-1 cells with control siRNA or *BAF155/170* siRNA (Fig 8C and 8D). We observed increased activity of the *Tead*-luciferase construct when transfected with Yap expression

plasmids. However, the increased *Tead*-luciferase reporter activity in response to Yap was significantly compromised in *BAF155/170* knockdown O9-1 cells compared to the control siRNA transfected O9-1 cells (Fig 8D). To further explore the Brg1-Tead interaction, we performed a luciferase assay and observed significant activation of the *Tead*-luciferase construct with Brg1. However, this activation was reduced by co-transfection of a dominant-negative form of Tead1 (DN-Tead1) (Fig 8E). We also observed that Brg1 and Yap synergistically activated the *Tead*-luciferase reporter (Fig 8F). However, this synergistic activation was reduced by co-transfection of a dominant-negative form of DN-Tead1 suggesting that Brg1-Yap-Tead interaction is essential for regulating the target gene expression (Fig 8G).

Next, we decided to perform transcription factor enrichment analysis as the BAF complex interact with diverse transcription factors to regulate gene expression and biological processes including embryonic development. To determine the enrichment of transcription factors targets in the list of significantly down-regulated genes in *BAF155/170*-deficient (*Wnt1$^{Cre/+}$; BAF155$^{fl/fl}$;BAF170$^{fl/fl}$*) NCCs, we used three different tools, namely Enrichr, PSCAN, and iCisTarget. As each of these methods is likely to contain some false-positives, we compared the results from the multiple methods via Venn analysis to identify robust findings across the methods employed (Fig 9A and S2 Table). We observed that many transcription factors identified from at least two out of three methods employed have been implicated in regulating the development of neural crest or neural crest-derived tissues. For example, we found enrichment in transcription factors that belong to the Notch family (Hey1, Hey2, and Hes5), AP-2 family (Tfap2a, Tfap2b, and Tfap2c), Myc family (Myc and MycN), F2F family (E2F1, E2F4, E2F7, and E2F8), Sox family (Sox1, Sox3, Sox4, Sox5, Sox6, Sox8, Sox9, Sox10, Sox11, Sox12, Sox13, Sox17, and Sox21) and Pbx family (Pbx1 and Pbx3) (Fig 9A).

To determine which cell population is most severely affected due to *BAF155/170* deletion, we compared the genes that are differentially expressed in our RNAseq analysis with a recently published neural crest single-cell RNAseq (scRNAseq) study [29]. Specifically, we examined the enrichment of differentially expressed genes (DEGs) observed in our study among the various expression-based cell clusters reported from the scRNAseq study, via Fisher's exact test. We observed that the majority of DEGs in *BAF155/170*-deficient (*Wnt1$^{Cre/+}$;BAF155$^{fl/fl}$; BAF170$^{fl/fl}$*) NCCs belong to mesenchymal, sensory, pre-delaminatory, and neural tube clusters. Among the significant clusters, the strongest enrichment was observed for the mesenchymal cluster (p = 3.19E-08) and the weakest for the neural tube cluster (p = 0.0039) (Fig 9B). We complemented this analysis by performing GSEA on our gene expression data, but after incorporating the scRNAseq reported gene clusters as 'custom' pathways on a background of hallmark gene sets from Molecular Signatures Database (MSigDB). Six of the seven custom pathways were retrieved as top GSEA hits (Fig 9C and 9D), with pathways that are significantly affected in the Fisher test occupying higher ranks in the GSEA output when ranked by normalized enrichment scores (NES) (Fig 9C and 9D). To identify the cell population that is most severely affected due to inactivation of Brg1-Yap-Tead interaction, we analyzed the Tead binding motifs in the 5kb promoter of DEGs identified in *BAF155/170*-deficient NCCs (S10A Fig). We observed a significant number of DEGs with Tead motifs. Next, we determined the number of Tead motifs in each DEGs and found that over a hundred genes with more than one and less than five Tead motifs suggesting that they are potential Brg1-Yap-Tead targets (S10B Fig). To determine the most severely affected population, we compared these DEGs with the neural crest scRNAseq data set. We observed that the majority of these DEGs belong to mesenchymal, sensory, pre-delaminatory, and neural tube clusters suggesting that proliferation, differentiation, and/or survival of these clusters are dependent on Brg1-Yap-Tead interaction (S10C Fig). These findings suggest that interaction between the BAF complex and transcription factors are essential for the development of neural crest-derived tissues.

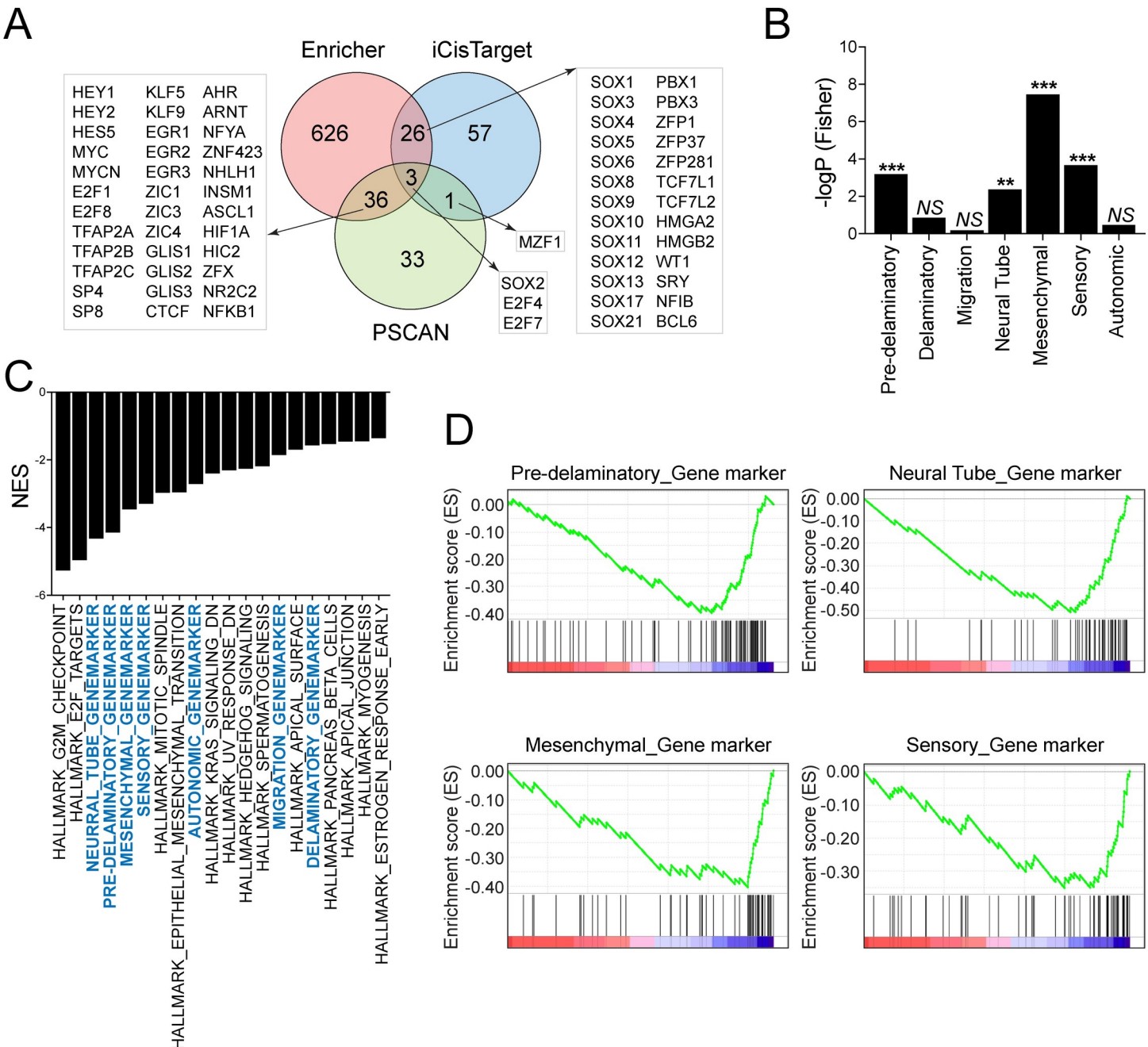

**Fig 9. Altered transcriptional network due to *BAF155/170* deletion in NCCs.** (**A**) Venn diagram on transcription factors which show significant enrichments in the *BAF155/170*-deficient (*Wnt1^Cre/+^;BAF155^fl/fl^;BAF170^fl/fl^*) NCCs compared to the controls. RNAseq data were analyzed via Enrichr, Pscan, and iCisTarget. (**B**) Fisher's test showing enrichment of differentially expressed neural crest genes in scRNAseq clusters identified by Soldato et al., 2019. (**C**) GSEA analysis showed that custom pathways (with cluster genes identified by Soldato et al., 2019, highlighted with blue color) are retrieved as top GSEA hits (top 20 pathways are presented). Pathways that are significant in the Fisher test occupy a higher rank in the GSEA output. (**D**) GSEA was performed to analyze the custom pathways affected by BAF155/170 deletion in NCCs.

## Discussion

Neural crest development is an extraordinarily complex process as NCCs give rise to many different cell types and contribute to a striking number of tissues and organs in the developing embryo. During this process, the gene expression programs in the NCCs are tightly regulated

so that they can delaminate, migrate to an array of tissues, and proliferate and differentiate into various lineages. Epigenetic modulators play crucial roles in regulating different aspects of neural crest development by modulating gene expression through DNA modifications, histone modifications, or by altering chromatin architecture. Here we demonstrate that the ATP-dependent chromatin remodeling complex BAF (mammalian SWI/SNF complex) is crucial for modulating the gene regulatory network, required for the proper migration, survival, proliferation, and differentiation of NCCs. The BAF complex contains up to 15 subunits, including two subunits with catalytic ATPase activity (Brg1 or Brm). The ATPase subunits are bound to the core subunits (BAF47, BAF155, and BAF170), and a variety of lineage-restricted subunits. Previous reports have shown that the deletion of a single BAF subunit does not affect the expression and incorporation of other subunits into Brg1 or Brm-based complexes suggesting that a single subunit deletion may not provide a complete understanding of the role of the BAF complex in neural crest development. For example, neural crest-specific deletion of *Brg1* leads to early embryonic death at E11.5 due to cerebral hemorrhage, abnormal pharyngeal arch artery patterning and remodeling, and cardiac OFT defects [14]. In contrast, the deletion of BAF complex subunit *BAF250a* in NCCs leads to partial lethality as 50% of embryos can survive between E15.5-P0. Even though the two mutants displayed similar phenotypes such as OFT defects, the molecular changes observed were different [15].

Recent studies have shown that dimerization of BAF155 and BAF170 subunits is the first step of the complex assembly and essential for the stability and functions of the BAF complex [4,13]. Therefore, to determine the role of the BAF complex in NCCs, we decided to delete the *BAF155* and *BAF170* subunits using $Pax3^{Cre/+}$ and $Wnt1^{Cre/+}$ lines. We observed that both $Wnt1^{Cre/+};BAF170^{fl/fl};BAF155^{fl/fl}$ and $Pax3^{Cre/+};BAF170^{fl/fl};BAF155^{fl/fl}$ embryos die between E11.5–12.5 due to various developmental defects including craniofacial defects, forebrain hemorrhage, pharyngeal arch artery defects, cardiac OFT defects, etc. Consistent with the OFT defects seen in neural crest-specific *Brg1* and *BAF250a* mutants, the depth of NCCs penetration into the heart was significantly reduced in $Wnt1^{Cre/+};BAF170^{fl/fl};BAF155^{fl/fl}$ embryos. *PlexinA2*, which encodes a semaphorin signaling receptor, is expressed in the pre-migratory NCCs in the dorsal neural tube and required for the proper guidance of cardiac NCCs into the proximal OFT [30]. In NCCs, Brg1 directly binds to *the PlexinA2* promoter and modulate its expression. In *Brg1* mutants, *PlexinA2* expression is significantly reduced in the NCCs as well as in the OFT [14]. Similar to *Brg1* mutants, PlexinA2 expression was significantly reduced in the OFT as well as in the sorted NCCs of the $Wnt1^{Cre/+};BAF170^{fl/fl};BAF155^{fl/fl}$ embryos compared to the controls. The OFT defect and molecular changes observed in $Wnt1^{Cre/+};BAF170^{fl/fl};BAF155^{fl/fl}$ mutants are very similar to the *Brg1* mutants suggesting that the BAF155 and BAF170 subunits as essential regulators of BAF complex functions [14]. However, despite similar OFT defects, *PlexinA2* expression was not altered in *BAF250a* mutants, suggesting that distinct BAF complexes may regulate different aspects of neural crest development [15]. Consistent with these findings, several studies using exome sequencing have demonstrated the association between *de novo* mutation in the BAF subunits with a range of human congenital and adult diseases including cardiac diseases [31–34].

Phenotypic analysis of *BAF155*-deficient ($Wnt1^{Cre/+};BAF155^{fl/fl};BAF170^{fl/+}$ and $Pax3^{Cre/+};BAF155^{fl/fl};BAF170^{fl/+}$) and *BAF170*-deficient ($Wnt1^{Cre/+};BAF170^{fl/fl};BAF155^{fl/+}$ and $Pax3^{Cre/+};BAF170^{fl/fl};BAF155^{fl/+}$) embryos suggest that compared to BAF170, BAF155 has a more significant role to play in neural crest progenitors, possibly due to the differences in their expression levels. These findings are consistent with previous observations that BAF155 is strongly expressed in the progenitor cells and its expression is gradually downregulated as differentiation proceeds [10–12,35]. In contrast, BAF170 is weakly expressed in progenitor cells and strongly expressed in differentiated cells [10–12,35]. It has been shown that paralogous

subunits occupy a similar key complex position, have partially redundant functions, and can compensate for each other to some extent. For example, loss of Brg1 increases the cell's dependency on Brm, evident by greater incorporation of Brm subunit into the BAF complex [36,37]. Similarly, BAF250b plays an essential role in stabilizing the BAF complex and preserving its functions in the absence of BAF250a [38–40]. In most cases, the compensatory effects are mutual and do not depend upon which subunit is targeted. However, in some cases, the compensatory effects are unidirectional [41]. Consistent with these findings, BAF155 may be able to compensate for the loss of BAF170 but BAF170 is not able to compensate for the loss of BAF155 in NCCs resulting in a more severe phenotype in *BAF155* mutants. Compared to the *BAF155*-deficient or *BAF170*-deficient embryos, a more severe phenotype in *BAF155/170*-deficient embryos indicates that both subunits are required for the formation, stability as well as functions of the BAF complex in NCCs.

Our data demonstrate that loss of *BAF155/170* leads to a significant increase in cell death and a decrease in cell proliferation and differentiation. Increased apoptosis and decreased cell proliferation were evident in the *BAF155/170*-deficient pharyngeal arches as well as around the neural tube. NCCs were able to migrate to pharyngeal arches but failed to differentiate into smooth muscle cells. Similarly, cardiac NCC differentiation into smooth muscle in the OFT was reduced. Instead of taking a candidate based approach used in previous studies, we performed RNAseq analysis on sorted NCCs to determine the global changes in gene expression due to *BAF155/170* deletion [14,15]. Gene expression changes identified by our RNAseq analysis were consistent with the cellular phenotype observed. Previous studies have shown that defects in multiple signaling pathways affecting cardiac NCC survival, migration, proliferation, or differentiation prevent normal development of the aortic arch arteries and cardiac OFT. Pathway enrichment analysis using RNAseq data showed that the Notch and Hippo signaling pathways are among the top-enriched pathways. The majority of the Notch signaling pathway genes such as *Jag2*, *Notch1*, *Notch3*, *Hes1*, *Hes5*, and *HeyL* are downregulated in *Wnt1^{Cre/+}; BAF170^{fl/fl};BAF155^{fl/fl}* NCCs. Our transcription factor enrichment analysis also showed enrichment in Hey1, Hey2, and Hes5 suggesting that their target genes are downregulated in *Wnt1^{Cre/+};BAF170^{fl/fl};BAF155^{fl/fl}* NCCs.

Previous studies have demonstrated that the Notch signaling pathway is essential for the migration, proliferation, as well as differentiation of the NCCs. For example, inhibition of Notch signaling in NCCs (by expressing a dominant-negative MAML or deleting *RBP-J*) results in cardiac OFT and aortic arch artery defects due to a significant reduction in the number of smooth muscle cells [42,43]. Deletion of *Notch3*, encoding a Notch receptor leads to reduced expression of vascular smooth muscle markers in a subset of arteries, suggesting that Notch signaling promotes smooth muscle differentiation or maturation [44]. Notch signaling has also been associated with apoptosis in smooth muscle cells and Notch3 specifically promotes smooth muscle cell survival [45–48]. Genetic mutations in *NOTCH3* are also associated with the autosomal dominant disorder CADASIL (cerebral autosomal dominant arteriopathy with subcortical infarcts and leukoencephalopathy) syndrome [49,50]. Neural crest-specific deletion of Hes1, one of the downstream targets of the Notch signaling pathway leads to impaired development of the pharyngeal organs due to defective proliferation and survival of neural-crest-derived mesenchymal cells in pharyngeal arches [51]. A comparison of our RNAseq data with the recently published single-cell sequencing data also demonstrated that the gene expression changes observed in the BAF mutants are most significantly enriched in the mesenchymal cell population [29,51]. Interactions between the BAF complex and Notch signaling components have been demonstrated in other biological contexts and are also consistent with our observation that the BAF complex contributes positively to the Notch-dependent transcriptional responses [52–54].

Another pathway that was significantly affected due to *BAF155/170* deletion was the Hippo signaling pathway. Expression of Hippo signaling pathway components such as *Yap*, *Tead1*, *Tead2*, *Tead3*, *Amot*, and *Axin2* was significantly decreased in *BAF155/170*-deficient (*Wnt1^Cre/+^;BAF155^fl/fl^;BAF170^fl/fl^*) NCCs. Hippo signaling pathway is a highly conserved pathway that regulates cellular proliferation, differentiation, and survival [55–61]. The downstream effector's Yap and Taz interact with transcription factors including Tead1-4 to promote migration, proliferation, and differentiation of NCCs in different biological systems. For example, Yap activity promotes human neural crest cell fate and migration *in vitro* [55]. In the avian embryo, Yap interacts with Wnt and BMP signaling pathway to regulate the proliferation and survival of pre-migratory NCCs [62]. Consistent with these findings, in addition to Yap, Wnt and BMP pathway components were also downregulated in *BAF155/170*-deficient NCCs. In mouse models, neural crest-specific deletion of Yap and Taz resulted in severe early embryonic lethality due to multiple defects including craniofacial defects, vascular defects due to impaired differentiation of NCCs into smooth muscle cells, forebrain hemorrhage, and neural tube defects [63–65]. Forebrain hemorrhage observed in *Yap/Taz* mutant embryos is similar to the one observed in Brg1 mutant (*Wnt1^Cre/+^;Brg1^fl/fl^*) or *BAF155/170* mutant (*Wnt1^Cre/+^;BAF170^fl/fl^;BAF155^fl/fl^*) embryos [14,63,65]. Yap and Taz not only promote cell proliferation and survival but also smooth muscle cell differentiation by modulating Notch signaling [64,65]. Yap and NICD (Notch intracellular domain) physically interact and regulate transcription in smooth muscle cells [64]. A recent study demonstrated that a shift in cellular metabolism promotes neural crest migration via Yap/Tead signaling [66]. Another study demonstrated that Yap interacts with BAF complex via BAF250a and depletion of BAF250a impaired the ability of Yap to be incorporated into the BAF complex [67]. Brg1 or Brm depletion did not affect the interaction between Yap and BAF250a. Canonically, Yap binds to Tead factors to induce transcription of target genes. However, no interaction between BAF250a and Tead was observed [67]. Here, we demonstrate that Brg1 physically interacts with the Yap and Tead transcription factor to regulate target gene expression. No change in Brg1-Yap interaction was observed due to either *BAF155* or *BAF155/170* knockdown. However, Brg1-Tead, as well as Yap-Tead interactions, were significantly compromised in both *BAF155* and *BAF155/170* knockdown cells suggesting that proper assembly of BAF complex is essential for Brg1-Yap-Tead-dependent transcription of target genes in NCCs.

Surprisingly, despite the phenotypic similarity, in contrast to the molecular changes observed in *Brg1* mutants such as increased levels of p21^cip1^ (*Cdkn1a*) and *Ask1* (apoptosis signal-regulating kinase 1), RNAseq analysis of sorted *BAF155/170*-deficient (*Wnt1^Cre/+^;BAF155^fl/fl^;BAF170^fl/fl^*) NCCs revealed no significant changes in the expression of p21^cip1^ or *Ask1* [14]. Instead, we observed reduced expression of p57^kip2^ (*Cdkn1c*), an inhibitor of several cyclin/Cdk complexes and a negative regulator of cell proliferation. We also observed elevated expression of genes involved in regulating apoptosis such as *Bcl10*, *Fas*, etc. In addition to Notch and Hippo signaling pathways, many other signaling pathways as well as transcription factors were significantly downregulated and may contribute to the defects observed in *BAF155/170* mutants. Their role in BAF complex-dependent neural crest development needs to be further explored. In conclusion, we highlight the function of the BAF complex in neural crest development. Our study revealed a previously undescribed role of the BAF complex in regulating neural crest-specific gene programs required for the proper development of neural crest-derived tissues including craniofacial, pharyngeal arch arteries, and OFT. Given that *de novo* mutations in BAF subunits are associated with a range of human congenital and adult malformations and syndromes, our findings may be relevant and can provide a better understanding of these human diseases.

## Materials and methods

### Ethics statement

All animal procedures were approved by the Institutional Animal Care and Use Committee at Duke-NUS Medical School/Singhealth conforming to the Guide for the Care and Use of Laboratory Animals (National Academies Press, 2011).

### Animal experiments

*BAF155/170* mutant mice were generated by crossing $Wnt1^{Cre/+}$ or $Pax3^{Cre/+}$ transgenic mice with $BAF155^{flox/flox};BAF170^{flox/flox}$ mice [13, 16, 17, 20]. Generated $Wnt1^{Cre/+};BAF155^{flox/+};$ $BAF170^{flox/+}$ or $Pax3^{Cre/+};BAF155^{flox/+};BAF170^{flox/+}$ were then backcrossed to $BAF155^{flox/flox};$ $BAF170^{flox/flox}$ mice to obtain *BAF155*-deficient ($Pax3^{Cre/+};BAF155^{flox/flox};BAF170^{flox/+}$ or $Wnt1^{Cre/+};BAF155^{flox/flox};BAF170^{flox/+}$), *BAF170*-deficient ($Pax3^{Cre/+};BAF155^{flox/+};BAF170^{flox/flox}$ or $Wnt1^{Cre/+};BAF155^{flox/+};BAF170^{flox/flox}$) and *BAF155/170*-deficient ($Pax3^{Cre/+};BAF155^{flox/flox};BAF170^{flox/flox}$ or $Wnt1^{Cre/+};BAF155^{flox/flox};BAF170^{flox/flox}$) embryos. Embryos were harvested from timed pregnancies counting the afternoon of the plug date as E0.5. Embryos were dissected in PBS and fixed in 4% paraformaldehyde (PFA) solution in PBS. Genotyping was performed on DNA isolated from either yolk sacs or tail biopsies using following primers: 5'-ATTCTCCCACCGTCAGTACG-3' and 5'-CGTTTTCTGAGCATACCTGGA-3' for $Pax3^{Cre/+}$; 5'-CAG CGC CGC AAC TAT AAG AG-3' and 5'-CAT CGA CCG GTA ATG CAG-3' for $Wnt1^{Cre/+}$; 5'-TGTCATCCATGAGGAGTGGTC-3' and 5'-GGTAGCTCACAA ATGCCTG T-3' for $BAF155^{flox/flox}$ and 5'-GATGCCTGCTTGCCGAATATCATG-3', 5'-CATGGTGG CTCTCCTAAGCAATCCAA-3' and 5'-CTGGCTTTGTGTGTGTGTGTTTGTTC-3' for $BAF170^{flox/flox}$. $R26^{mTmG/+}$ embryos were genotyped based on RFP expression.

### Histology and immunostaining

Histology and immunostaining were performed as described previously [68–71]. Briefly, embryos harvested from timed-mating were fixed with 4% formaldehyde overnight and subsequently dehydrated in ethanol and embedded in paraffin for sectioning. Immunofluorescence staining was carried out by dewaxing of the sections followed by antigen retrieval, rehydration, and blocking with 5% BSA-PBST (0.1%) for 2–3 hours at room temperature. Next, sections were incubated with primary antibodies overnight at 4°C, followed by washing and secondary antibody incubation for 1–2 hrs at room temperature the next day. Antibodies used include anti-GFP (1:250 dilution, Cell Signaling, Catalog no. 2956S), anti-SM22α (1:100 dilution, Abcam, Catalog. no. ab14106), and anti-PlexinA2 antibody (1:100 dilution, Abcam, Catalog. no. ab39357). Whole-mount immunostaining for neurofilament (2H3) was carried out as described previously [68,71–75].

### Cell proliferation and TUNEL assay

Cell proliferation in the neural tube and pharyngeal arches was evaluated by Ki67 immunostaining using anti-Ki67 (1:200 dilution, Abcam, Catalog. no. ab16667) antibody on E10.5 control and BAF155/170 double knockout sections. Dapi counterstain (Vector Laboratories) was performed to visualize the nuclei. Quantification of cell proliferation was calculated as the ratio of Ki67-positive cells to the total number of cells as determined by Dapi counterstaining in the defined area of the neural tube and pharyngeal arch. TUNEL assay (Sigma, Catalog no. 11684795910) was carried out with the protocol provided by the manufacturer. For each genotype, images of 4–6 different sections of 3–4 independent embryos were used.

## Neural crest cells isolation using fluorescence-activated cell sorting (FACS)

We performed timed-mating between $Wnt1^{Cre/+};BAF155^{flox/+};BAF170^{flox/+}$ and $BAF155^{flox/flox}$; $BAF170^{flox/flox};Rosa26^{mTmG/mTmG}$ mice to generate E11.25 control ($Wnt1^{Cre/+};BAF155^{flox/+};$ $BAF170^{flox/+};Rosa26^{mTmG/+}$) and $BAF155/170$-deficient ($Wnt1^{Cre/+};BAF155^{flox/flox};BAF170^{flox/flox};Rosa26^{mTmG/+}$) embryos for FACS. Embryos were dissected in cold DPBS and subsequently incubated in 0.5% trypsin-EDTA (without phenol red) at 37°C for 20 minutes. Homogenized tissue was centrifuged at 2000 rpm for 5 minutes. Cell pellet was washed with 5% FBS/DPBS, centrifuged, and re-suspended with 5% FBS/DPBS. The cell suspension was transferred to a clean tube with cell-strainer cap, and used for FACS sorting. GFP⁻ ($BAF155^{flox/+};BAF170^{flox/+};$ $Rosa26^{mTmG/+}$) embryos were used as a negative control for the FACS. GFP⁺ neural crest cells from 3 control ($Wnt1^{Cre/+};BAF155^{flox/+};BAF170^{flox/+};Rosa26^{mTmG/+}$) and 3 $BAF155/170$- deficient ($Wnt1^{Cre/+};BAF155^{flox/flox};BAF170^{flox/flox};Rosa26^{mTmG/+}$) embryos were collected.

## RNA extraction, library preparation, and RNASeq analysis

RNA from collected neural crest cells was extracted using FavorPre RNA isolation kit (Catalog no. FATRK001). Subsequently, strand-specific poly(A) RNAseq libraries were generated using NEBNext. Ultra II Directional RNA Library Prep Kit with sample purification beads from Illumina (NEB, Catalog no. E7795S). Each sample is individually barcoded and pooled for paired-end sequencing using the Illumina HiSeq4000 platform at Novogene AIT Genomics Singapore.

Paired-end RNA sequencing reads from each sample (median read depth > 32 million) were adapter-trimmed (to <10% adapter sequences in a read) and aligned to the mouse reference genome (GRCm38) via STAR aligner (version 2.5.1b) [76] with a median mapping rate of 68 percent. Mapped sequencing reads were quantified and assigned to genomic features via the *feature count* option in R package *Rsubread* [77]. Differential gene expression analysis was conducted in R via *limma* [78]. Genes with absolute log fold change ≥0.58, average signal (counts per million)> = 2, and adjusted p-value (FDR) ≤ 0.1 were considered to be differentially expressed. Transcriptome data were further analyzed for enrichment of biological pathways by querying 'gene-sets' from the Kyoto Encyclopedia of Genes and Genomes (KEGG) [79] or Gene Ontology [80] pathway repositories via the PreRankedGSEA [81] or Enrichr over-representation analysis tools [82]. Prediction of candidate upstream regulators such as transcription factors was carried out on the list of differentially expressed genes via three independent approaches (Enrichr, PSCAN [83] and iCisTarget [84]. For Enrichr, candidate transcription factors were selected based on four different data-sources, Chea, Encode-Chea, ARCHS4, and TFperturb). PSCAN analysis was conducted based on position weight matrices (PWMs) from the Jaspar 2018_NR dataset. iCisTarget analysis was conducted through an analysis of over-representation in PWMs and TF-binding sites from publicly available Chipseq data. A normalized enrichment score >3 was considered as the lower bound for enrichment. The performance of each method for the identification of candidate transcription factors was assessed by Venn analysis.

Gene markers of single-cell sequencing unbiased clusters of trunk E9.5 neural crest were downloaded from the published paper [29]. The gene marker clusters are neural tube (n = 67), pre-delaminatory (n = 109), delaminatory (n = 27), migration (n = 19), sensory (n = 77), autonomic (n = 41) and mesenchymal (n = 60). Fisher exact test (R programming; URL-https://www.r-project.org/) was deployed to look at the enrichment of $BAF155/170$-neural crest differentially expressed genes (DEGs) in single-cell sequencing clusters (Fisher's test). DEGs are significantly enriched in the neural tube, pre-delaminatory, sensory, and mesenchymal gene clusters. The mesenchymal gene cluster is most enriched for differentially expressed genes.

GSEA analysis was carried out on the full list of genes from *BAF155/170*-neural crest cells RNAseq data, ranked by their fold-change (Double mutant vs Control) [85]. Cluster genes were included as 'custom' pathways on a background of Hallmark genesets from MSigDB.

## RNA isolation, cDNA synthesis, and qRT-PCR

Total RNA from sorted control and BAF155/170 knockout NCCs was isolated using Favorgen RNA extraction kit (Favorgen Biotech, Catalog no. FATRK 001–2). The concentration and quality of RNA were measured using NanoDrop. For cDNA synthesis, total RNA was reverse transcribed using random hexamers and Promega M-MLV reverse transcriptase (Promega, catalog no. M1705). Gene expression analysis was performed by quantitative RT-PCR (ABI PRISM 7900 or ViiA7 Real-Time PCR System) using SYBR Green Master Mix (Life Technologies, catalog no. 4368702). All mRNA data were normalized against the reference gene glyceraldehyde-3 phosphate dehydrogenase (Gapdh). Primers used for RT-PCR analysis are listed below.

Jagged2 F: 5'-GTG GCT GTG TCT TTC AGC CC-3'
Jagged2 R: 5'-CAC AGC ACG GGC ACC AAC AG-3'
NOTCH1 F: 5'-GAT GAC CTA GGC AAG TCG GC-3'
NOTCH1 R: 5'-GCA GTC TCA TAG CTG CCC TCA C-3'
NOTCH3 F: 5'-GAT GAG CTT GGG AAA TCT GCC-3'
NOTCH3 R: 5'-GCT TGG CAG CCT CAT AGC TG-3'
Hes1 F: 5'-CCAGCCAGTGTCAACACGA-3'
Hes1 R: 5'-AATGCCGGGAGCTATCTTTCT-3'
Hes5 F: 5'-AGTCCCAAGGAGAAAAACCGA-3'
Hes5 R: 5'-GCTGTGTTTCAGGTAGCTGAC-3'
Heyl F: 5'-GAG AAA CAG GGC TCC TCC AAG-3'
Heyl R: 5'-GAC CTC AGT GAG GCA TTC CC-3'
Yap1 F: 5'-TGG CCA AGA CAT CTT CTG GT-3'
Yap1 R: 5'-GCC ATG TTG TTG TCT GAT CG-3'
Tead1 F: 5'-CTG GCC TGG GAT GAT ACA GAC-3'
Tead1 R: 5'-GCT GAC GTA GGC TCA AAC CC-3'
Tead2 F: 5'-CCA GAC GTG AAG CCC TTC TC-3'
Tead2 R: 5'-TAC CCT GGT AGG TCA GAG GC-3'
Tead3 F: 5'-GCA TTA AGG CTA TGA ACC TGG AC-3'
Tead3 R: 5'-CAG AGA CGA TTT GGG CAG AC-3'
Amot F: 5'-CCGCCAGAATACCCTTTCAAG-3'
Amot R: 5'-CTCATCAGTTGCCCCTCTGT-3'
Axin2 F: 5'-TGACTCTCCTTCCAGATCCCA-3'
Axin2 R: 5'-TGCCCACACTAGGCTGACA-3'
plexinA2 F: 5'-AGT GAC CAA GAT ATG AAT GCC TCA CT-3'
plexinA2 R: 5'-TCT ATG GAC ATG GCG TTG ATG-3'

## Cell culture and siRNA knockdown

O9-1 cells (Merck, Catalog no. SCC049) were cultured in complete ES cell medium with 15% FBS and LIF (Merck, Catalog no. ES101-B) supplemented with 25ng/mL of recombinant human FGF-basic protein (Fisher scientific, GF003). Cells are cultured on plates coated with Geltrex LDEV-Free Reduced Growth Factor Basement Membrane Matrix (Thermo Fisher Scientific, A1413201) at 37°C, 5% $CO_2$. The cell medium is changed every 2–3 days. For passaging, confluent cells are washed with DPBS and treated with 0.25% trypsin-EDTA at 37°C for

3–5 minutes, neutralized with 10% FBS in DMEM, and centrifuged at 300rpm, 4°C for 5 minutes. Cells are resuspended in pre-warmed medium and seeded to new plates with a ratio of 1:4. Control (Thermo Fisher Scientific, catalog no. 4390843), *Smarcc1* (*BAF155*) siRNA (Thermo Fisher Scientific, catalog no. AM16708), and *Smarcc2* (*BAF170*) siRNA (Thermo Fisher Scientific, catalog no. 4390771) were obtained. The siRNA was transfected in O9-1 cells using Lipofectamine RNAiMAX transfection reagent (Thermo Fisher Scientific, catalog no. 13778150). Cell extracts were prepared at 72 hrs post-transfection using RIPA buffer (Thermo Fisher Scientific, catalog no. 89901) for western blot analysis to evaluate the knockdown efficiency. Western blots were performed as described previously [68–71].

## Co-immunoprecipitation (Co-IP), and western blot

Cell extracts from O9-1 cells transfected with control or *Smarcc1* (*BAF155*)/*Smarcc2* (*BAF170*) mix siRNA pool were prepared using cold RIPA buffer (Thermo Fisher Scientific, catalog no. 89901) supplemented with Halt protease and phosphatase inhibitor cocktail (100X) (Thermo Fisher Scientific, catalog no. 78440). Protein concentration was checked with spectrophotometry as compared to standard BSA solutions. For the co-immunoprecipitation experiment, cell extracts were then incubated with control IgG, anti-Tead antibody (1:100 dilution; Cell Signaling, Catalog no. 13295S) or anti-Brg1 antibody (1:100 dilution, Abcam, Catalog no. ab-70558) at 4°C overnight. The next day, 20–30μL of protein A-agarose beads (Santa Cruz, Catalog no. sc-2001) were added into the cells extracts and further incubated at 4°C for 2 hours on a roller. After a quick spin, the supernatant was removed and agarose beads were washed thrice with 1mL of cold PBS supplemented with inhibitors. Finally, immunoprecipitated proteins were extracted in Laemmli sample buffer (BIO-RAD, Catalog no.1610737) and evaluated with western blot analysis. Input western blots were also performed with 10% of the total protein used for the Co-IP experiment. Western blotting experiments were performed as described previously [68–71]. The primary antibodies used include anti-BAF155 (1:300 dilution, Santa Cruz, Catalog no. sc-32763), anti-BAF170 (1:300 dilution, Santa Cruz, Catalog no. sc-17838), anti-β-actin (1:1000 dilution, Santa Cruz, Catalog no. sc-47778), anti-Brg1 (1:1000 dilution, Abcam, Catalog no. ab-70558), anti-Tead (1:1000 dilution, Cell Signaling, Catalog no. 13295S) and anti-Yap (1:300 dilution, Santa Cruz, Catalog no. sc-376830). The secondary antibodies used include rabbit anti-mouse IgG H&L (HRP) (1:5000 dilution, Abcam, Catalog no. ab97046) and goat anti-rabbit IgG H&L (HRP) (1:5000 dilution, Abcam, Catalog no. ab6721).

## Luciferase assay

Luciferase assay was performed as previously described [68,71,86]. Briefly, O9-1 cells were cultured as described above and transfected with control or *BAF155*/*BAF170* siRNA mix using Lipofectamine RNAiMAX transfection reagent (Thermo Fisher, Catalog no. 13778150). After 24 hrs of siRNA transfection, O9-1 cells were re-transfected with Hes1 or Tead-luciferase reporter plasmid together with plasmids expressing NICD or Yap respectively. To normalize transfection, β-galactosidase expression plasmid was also transfected along with other plasmids. After 48 hours, cell extracts were prepared using lysis buffer (Promega, catalog no. E3971). Next, luciferase activities were measured using the Luciferase Reporter Assay System Kit (Promega, catalog no. E1500) and β-galactosidase activity using the β-Galactosidase Enzyme Assay System (Promega, catalog no. E2000). The luciferase reporter activity is normalized to β-galactosidase activity. Experiments were repeated in at least three independent experiments and duplicates. Plasmids used in luciferase experiments include pHes1 (2.5k)-luc, a gift from Ryoichiro Kageyama (Addgene plasmid 43806) and 3XFlagNICD1 is a gift from Raphael Kopan (Addgene plasmid 20183) [87,88], WT-Smarca4-sfGFP is a gift from Jerry Crabtree & Courtney

Hodges (Addgene plasmid 107056) [89]. Yap expression plasmid has been previously described [68,90] and 8xGTIIC-luciferase, a gift from Stefano Piccolo (Addgene plasmid 34615) [91].

## Statistical analysis

Statistical analyses were performed using the two-tailed Student's t-test. Data are expressed as mean ± standard error mean (SEM). Differences were considered significant when the p-value was <0.05. (*, $p<0.05$; **, $p<0.01$; ***, $p<0.001$; NS, not significant). Statistical analyses were performed using Graph-Pad Prism v.5 (GraphPad Software, USA).

## Supporting information

**S1 Fig. BAF155 and BAF170 expression in neural crest cells.** (A-B) Double immunostaining for BAF155 and Pax3 was performed on E10.5 transverse sections. (C-D) Double immunostaining for BAF170 and Pax3 was performed on E10.5 transverse sections. Scale bars 100μM. NT, Neural tube, DRG, dorsal root ganglion.
(TIF)

**S2 Fig. Impaired differentiation of NCCs into smooth muscle cells of the pharyngeal arch arteries.** (A-D) Immunostaining for SM22α on the frontal section from E11.5 control, *BAF170*-deficient (*Pax3$^{Cre/+}$;BAF170$^{fl/fl}$;BAF155$^{fl/+}$*), BAF155-deficient (*Pax3$^{Cre/+}$;BAF155$^{fl/fl}$; BAF170$^{fl/+}$*) and *BAF155/170*-deficient (*Pax3$^{Cre/+}$;BAF155$^{fl/fl}$;BAF170$^{fl/fl}$*) embryos (n = 3–4 for each genotype). Nuclei were visualized by Dapi staining (blue). 4$^{th}$ Pharyngeal arch artery (paa) is shown. Scale bar 50μM (A-D).
(TIF)

**S3 Fig. Cleft palate in *BAF170*-deficient embryos.** (A-F) To analyze the palates, E15.5 control (A-C, n = 4) and *BAF170*-deficient (*Wnt1$^{Cre/+}$;BAF170$^{fl/fl}$;BAF155$^{fl/+}$*) embryos (D-F, n = 4) were harvested. No obvious defects were observed by whole embryos morphology (A and D). Compared to the controls, palates were not fused in the *BAF170*-deficient (*Wnt1$^{Cre/+}$; BAF170$^{fl/fl}$;BAF155$^{fl/+}$*) embryos (B and E). Alcian blue staining showed cleft palate in the *BAF170*-deficient (*Wnt1$^{Cre/+}$;BAF170$^{fl/fl}$;BAF155$^{fl/+}$*) embryos (C and F). (G-H) H&E stained palate sections of control (G) and *BAF170*-deficient (H) embryos. (I-K) Immunohistochemistry for Ki67 on palate sections of control (I) and *BAF170*-deficient (J) embryos and quantification (K). (L-N) TUNEL staining on palate sections of control (L) and *BAF170*-deficient (M) embryos and quantification (N). Nuclei were visualized by DAPI staining (blue). Asterisks (*) represent cleft palate in *BAF170*-deficient embryos (E, F and H). P, Palate; PS, Palatal shelves. Values are reported as means ± SEM (*P < 0.05, **P < 0.01, ***P < 0.001; NS, not significant). Scale bar 1mM (A and D). Scale bar 500μM (B-C and E-F). Scale bar 100μM (G-M).
(TIF)

**S4 Fig. Cleft palate defects in *BAF155* mutant (*BAF155$^{FoxG1-CKO}$*) embryos.** (A-L) Histological analysis of E15.5 control (A-F) and *BAF155$^{FoxG1-CKO}$* (*FoxG1$^{Cre/+}$;BAF155$^{fl/fl}$*) (G-L) embryos (n = 3 controls, n = 4 *BAF155$^{FoxG1-CKO}$*). Transverse sections of control and *BAF155$^{FoxG1-CKO}$* embryos at the level of the anterior to posterior palatal shelves were analyzed. Asterisks (*) represent cleft palate in *BAF155$^{FoxG1-CKO}$* embryos (J-L). PS, Palatal shelves; NS, Nasal septum; T, Tongue.
(TIF)

**S5 Fig. Impaired nervous system development in *BAF155/170*-deficient embryos.** (A-D) Whole-mount neurofilament (2H3) immunostaining of E10.5 control and *BAF155/170*-deficient (*Pax3$^{Cre/+}$;BAF155$^{fl/fl}$;BAF170$^{fl/fl}$*) embryos (n = 3 for each genotype). Impaired

development of the cranial nerves (V: trigeminal, VII/VIII: facial/vestibulocochlear, X: vagal) is highlighted. Scale bars 200μM.
(TIF)

**S6 Fig. Fluorescence-activated cell sorting (FACS) of GFP⁺ NCCs.** (A-B) Experimental design for NCCs isolation for RNAseq analysis using FACS. To setup the FACS, single cell suspensions were prepared from lineage traced GFP⁻ control ($BAF155^{fl/+}$:$BAF170^{fl/+}$:$Rosa26^{mTmG/+}$), GFP⁺ double heterozygous controls ($Wnt1^{Cre/+}$:$BAF155^{fl/+}$:$BAF170^{fl/+}$:$Rosa26^{mTmG/+}$) and $BAF155/170$-deficient ($Wnt1^{Cre/+}$:$BAF155^{fl/fl}$:$BAF170^{fl/fl}$:$Rosa26^{mTmG/+}$) embryos for FACS. NCCs that were positive for GFP were collected from double heterozygous controls ($Wnt1^{Cre/+}$:$BAF155^{fl/+}$:$BAF170^{fl/+}$:$Rosa26^{mTmG/+}$) and $BAF155/170$-deficient ($Wnt1^{Cre/+}$:$BAF155^{fl/fl}$:$BAF170^{fl/fl}$:$Rosa26^{mTmG/+}$) embryos for RNA isolation and subsequently for library preparation and RNAseq analysis.
(TIF)

**S7 Fig. Altered signaling pathways regulating NCCs migration, proliferation, and differentiation in $BAF155/170$ mutants.** GSEA analysis showing changes in signaling pathways known to regulate different aspects of neural crest development.
(TIF)

**S8 Fig. Tead expression in neural crest cells.** Immunostaining for Pan-Tead was performed on E10.5 transverse sections. Scale bars 100μM. NT, Neural tube.
(TIF)

**S9 Fig. BAF complex is essential for Notch target gene expression.** Results of normalized luciferase reporter assays in O9-1 cells with Hes1-luciferase reporters in the presence of control siRNA or $BAF155/170$ siRNA. Values are reported as means ± SD (*P < 0.05, **P < 0.01, ***P < 0.001; NS, not significant).
(TIF)

**S10 Fig. Neural crest cell population that is most severely affected due to inactivation of Brg1-Yap-Tead interaction.** (A) Tead binding motifs in the 5kb promoter of differentially expressed genes (DEGs) identified in $BAF155/170$-deficient NCCs. (B) Number of Tead motifs in the gene promoter of DEGs. Over 100 DEGs have more than one and less than five Tead motifs. (C) Fisher's test showing enrichment of differentially expressed neural crest genes with Tead motifs in scRNAseq clusters identified by Soldato et al., 2019.
(TIF)

**S1 Table. List of differentially expressed genes between control and BAF155/170 double mutant NCCs.**
(PDF)

**S2 Table. Transcription factor enrichment analysis using RNAseq data set.**
(PDF)

## Acknowledgments

We are thankful to M.K.S. lab members and Dr. Lena Ho (Duke-NUS Medical School) for helpful discussion.

## Author Contributions

**Conceptualization:** Manvendra K. Singh.

**Data curation:** Kathleen Wung Bi-Lin, Pratap Veerabrahma Seshachalam, Tran Tuoc, Anastassia Stoykova, Sujoy Ghosh, Manvendra K. Singh.

**Formal analysis:** Kathleen Wung Bi-Lin, Pratap Veerabrahma Seshachalam, Tran Tuoc, Sujoy Ghosh, Manvendra K. Singh.

**Funding acquisition:** Manvendra K. Singh.

**Investigation:** Kathleen Wung Bi-Lin, Pratap Veerabrahma Seshachalam, Tran Tuoc, Sujoy Ghosh, Manvendra K. Singh.

**Methodology:** Kathleen Wung Bi-Lin, Pratap Veerabrahma Seshachalam, Tran Tuoc, Sujoy Ghosh, Manvendra K. Singh.

**Project administration:** Manvendra K. Singh.

**Resources:** Tran Tuoc, Manvendra K. Singh.

**Software:** Kathleen Wung Bi-Lin, Pratap Veerabrahma Seshachalam, Sujoy Ghosh.

**Supervision:** Manvendra K. Singh.

**Validation:** Kathleen Wung Bi-Lin, Sujoy Ghosh.

**Visualization:** Kathleen Wung Bi-Lin, Tran Tuoc.

**Writing – original draft:** Kathleen Wung Bi-Lin, Manvendra K. Singh.

**Writing – review & editing:** Manvendra K. Singh.

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
