## [Decision Letter · Decision Letter 0]

4 Dec 2020

Dear Dr Singh,

Thank you very much for submitting your Research Article entitled 'Critical role of the BAF chromatin remodeling complex during murine neural crest development' to PLOS Genetics. Your manuscript was fully evaluated at the editorial level and by independent peer reviewers. The reviewers appreciated the attention to an important problem, but raised some substantial concerns about the current manuscript. Based on the reviews, we will not be able to accept this version of the manuscript, but we would be willing to review again a much-revised version. We cannot, of course, promise publication at that time.

If you decide to revise the manuscript for further consideration at PLOS Genetics, please aim to resubmit within the next 60 days, unless it will take extra time to address the concerns of the reviewers, in which case we would appreciate an expected resubmission date by email to plosgenetics@plos.org.

[LINK]

We are sorry that we cannot be more positive about your manuscript at this stage. Please do not hesitate to contact us if you have any concerns or questions.

Yours sincerely,

Frank L Conlon

Associate Editor

PLOS Genetics

Scott Williams

Section Editor: Natural Variation

PLOS Genetics

Though the reviewers found the topic and approach of interest and many findings novel, the reviewers raise several concerns that need to addressed. First, the phenotypic and molecular studies would need to be conducted at an earlier time point. Second, the molecular events need to be shown to be causative to the observed phenotypes and not a readout. Third, as pointed out in past reviews and, by two of the present reviewers the authors need to detail the aspects of neural crest cell biology is being affected, induction, specification, migration, differentiation. Lastly, the authors need to between neural crest and non-neural crest tissue in their RNA-seq data sets. We look forward to reviewing a revised version of the manuscript.

Reviewer's Responses to Questions

**Comments to the Authors:**

Reviewer #1: The BAF (mammalian SWI/SNF) complex is an ATP dependent chromatin remodeling complex. Composed of up to 15 subunits, the dimerization of BAF155 and BAF170 is thought to be the first step in complex assembly and be essential for the stability and function of the complex.

There is considerable interest in the factors that regulate neural crest cell development and this is true even for epigenetic regulators. In this regard the authors deleted BAF155 and BAF170 individually and in combination in neural crest cells using Pax3-Cre and Wnt1-Cre. Single BAF170 cko have a mild to no phenotype. In contrast single BAF155 CKO mice exhibit a stronger phenotype including cleft palate. However double BAF155/170 cko demonstrate the synergistic requirement for these sunubits as the double mutants are mid gestation embryo lethal and exhibit craniofacial anomalies such as cleft palate and cardiovascular anomalies including outfliow tract malformations. The authors make some interesting observations regarding altered Sma22 and PlexinA2 staining in the vasculature, 2H3 immunostaining of the the cranial ganglia and decreased NCC proliferation and increased apoptosis, and the failure of the palatal shelves to fuse resulting in cleft palate but I found these observations to be largely superficial.

A major weakness of the paper is that the craniofacial anomalies particularly with respect to the pharyngeal arches is evident in their smaller size at E10.5. The authors should have performed analyses of proliferation and apoptosis at E9.5 and if alterations were observed at that time point then they should have looked even earlier. It's clear that there is a deficiency in the number of migrating neural crest cells which on its own can explain the craniofacial anomalies including cleft palate.

In general I found the analyses of the distinct phenotypes a bit superficial. There was no examination of why the palatal shelves failed to fuse in the BAF155 CKO mice. It's not clear to me why the authors traced neural crest cells using mTmG and then immunostained for GFP with a red secondary fluorescent antibody as an indication of neural crest cell localization, nor why they chose to also reveal PlexinA2 immunostaining with red fluorescence. This was a missed opportunity to show co-localization of neural crest cells with the respective markers. It's important to demonstrate whether the absence of protein immunostaining is due to an absence of the neural crest cells in that same territory or an indication of a cell fate change. Nonetheless this was the extent of the cellular analyses in the cko mutants.

It's not clear to me why the authors chose E11.25 for their neural crest cell isolation and RNA-seq analyses. My concern is that many of the changes observed are likely secondary effects since they are observed well after the onset of the phenotype. But even if they could be classified as primary direct affects, the mechanism then needs to be evaluated in the mutant embryos. It doesn't make sense that the authors isolated neural crest cells at E11.25 and yet their data suggests pre-migratory neural crest factors are highly represented. The authors use luciferase and co-immunoprecipitation assays and competitive interactions assays in O9-1 cells to validate an interaction between BAF and Notch signaling, as well as BAF and HIPPO (Yap-Taz-Tead) signaling in vitro. While these interactions may be quite valid in neural crest cells at the E11.5 time point, the mutant embryo phenotype is apparent well before then. Thus there is a disconnect between the signaling links and their validation in a cell line with the temporal onset of the mutant embryo phenotype. As a result the authors rely heavily on assumed connections between these pathways from the literature and make correlative assertions about their importance in underlying the phenotypes but fail to demonstrate causation. For example a thorough examination of altered Notch siganling as a key driver of the cardiovasular defects should be demonstrated spatiotemporally in the mutants. Similarly HIPPO/Tead signaling should also be validated in the same way. Ultimately there was a missed opportunity to chemically stimulate Notch and HIPPO signaling to test if this could ameliorate the cardiovascular phenotype and in doing so validate the phenotypic connection to these pathways.

Other comments.

The authors state in the first line of the introduction that neural crest cells originate in the dorsal neural tube. This is incorrect with respect to cranial and cardiac neural crest cells in mice, which form and migrate from the dorsolateral edges of the neural plate prior to formation of the neural tube.

Reviewer #2: An interesting manuscript detailing the role of members of the BAF complex in neural crest development and downstream transcriptional consequences. Images are clear and denote profound phenotypes that are likely interpreted correctly. I applaud the extra work that was performed by the authors to satisfy previous reviewer concerns. While there are some minor spelling and grammatical issues, no major concerns are noted.

Reviewer #3: In the manuscript “Critical role of the BAF chromatin remodeling complex during murine neural crest development” Bin et al. use different Cre-lines (Pax3, Wnt1, FoxG1) to analyze the function of core subunits of the BAF complex (BAF155 and BAF170) in mouse development. They observe that neural crest-specific deletion of BAF155/170 causes defects in cranio-facial, pharyngeal arch artery, cardiac outflow tract development and results in embryonic lethality. Proliferation and apoptotic assays suggest that BAF155/170 is required for neural crest proliferation and survival. Furthermore, RNAseq analysis suggests that the BAF complex controls the expression of signaling pathway genes critical for neural crest proliferation, migration and differentiation, like regulators of Notch and Hippo signaling. The authors present interesting data suggesting that the BAF complex plays a role in neural crest development and they provide new insight that BAF155/170 is required for Brg1-Yap-Tead-dependent transcription of target genes. However, my concern is that they may be looking here at secondary effects:

1. The authors analyzed their Cre knockout lines at different time points (9.5 to 14.5). Fig. 1 and 2 starts by showing embryos at stage 10.5. Are there earlier defects in neural crest development? (The authors should also comment, why they choose to analyze embryos at stages 9.5 to 14.5.)

2. What is the temporal expression of BAF155/170 in particular at different stages of neural crest development (induction, specification, etc.)? They only briefly comment on BAF155/170 expression in progenitor cells versus differentiated cells in the introduction. However, it would be helpful to characterize the expression of BAF155/170 at different stages using IHC, like the authors have previously shown in the olfactory epithelium (Bachmann et al., 2016).

3. The authors performed RNAseq analysis of FACS-sorted neural crest cells of their Wnt1-cre lines. Why did they choose embryos of E11.25? Would it not be useful to look at earlier stages to determine when BAF155/170 activity is initially required? As they observe gene expression changes in neural crest specifiers (Fig. 6) this may indicate defects in neural crest specification. Thus, the observed effects on neural crest guidance cues, hippo signaling etc. may just be secondary to defects in neural crest specification.

4. As it is known that Wnt1-Cre labels a large population of non-neural crest cells in the neural plate this Cre-line may not be the best choice to determine gene expression changes in neural crest cells. In fact, they detect gene expression changes in a number of genes involved in axon guidance and neural crest migration, like different classes of semaphorins (Fig. 6). One would expect that these guidance cues are expressed in non-neural crest tissue, while neural crest cells express for example plexin receptors. Which leaves the question, what is the nature of the cells that they have been analyzing?

5. The authors use lineage-tracing to analyze if neural crest migration or differentiation are affected (Fig. 3). Although there are GFP-positive cells in the branchial arches of the BAF155/170Wnr1-CKO, the branchial arch area is dramatically smaller and there seem to be fewer migrating cells (Fig. 3D), which may indicate migration defects. The authors also show frontal sections, however, it is difficult to compare these; for example the controls seem to show less GFP signal. Better imaging and quantification is required to exclude migration defects.

6. In addition to point 5: The lack of SM2alpha (Fig. 3 M-P) or plexinA2-positive cells (Fig. 4 E-H) may simply result from defects in proliferation or increased apoptosis and not represent a defect in differentiation. For example the data in Fig. 3M-P could be interpreted as a differentiation defect, but may also result from defects in migration, proliferation or increased apoptosis.

7. Quantification of the relative distance of cardiac neural crest migration. How was the distance (arrowhead in Figs. 4) determined? From Figs. 4B,D it is not clear how they chose the length of the double head arrows.

**Have all data underlying the figures and results presented in the manuscript been provided?**

Reviewer #1: Yes

Reviewer #2: Yes

Reviewer #3: Yes

PLOS authors have the option to publish the peer review history of their article (what does this mean?). If published, this will include your full peer review and any attached files.

Reviewer #1: No

Reviewer #2: No

Reviewer #3: No

---

## [Decision Letter · Decision Letter 1]

19 Feb 2021

Dear Dr Singh,

Thank you very much for submitting your Research Article entitled 'Critical role of the BAF chromatin remodeling complex during murine neural crest development' to PLOS Genetics.

The manuscript was fully evaluated at the editorial level and by independent peer reviewers. The reviewers appreciated the attention to an important topic but Reviewer #1 identified some concerns that we ask you address in a revised manuscript

We therefore ask you to modify the manuscript according to the review recommendations. Your revisions should address the specific points made by each reviewer.

[LINK]

Yours sincerely,

Frank L Conlon

Associate Editor

PLOS Genetics

Scott Williams

Section Editor: Natural Variation

PLOS Genetics

Reviewer's Responses to Questions

**Comments to the Authors:**

Reviewer #1: The authors have made considerable revisions which have improved the manuscript. The new apoptotic data at E10.5 helps to explain the smaller pharyngeal arches in the mutants at this stage and prefaces the ensuing defects or agenesis of neural crest cell derivatives. The smaller pharyngeal arches are also evident in the 2H3 stained E10.5 embryos.

One of the surprising outcomes from the RNA-seq data and subsequent bioinformatic and pathway analyses is the presence of neural crest cell pre-EMT and delamination as major pathways involving differentially expressed genes. Considering the data was obtained from neural crest cells in E11.25 embryos there is a bit of a temporal disconnect since these events in neural crest development take place from E8.5-9.5 and the BAF155/170 mutants don't exhibit an EMT/delamination failure phenotype. The authors should at least comment on this temporal developmental discrepancy as well as the phenotypic discrepancy. The phenotypic discrepancy calls to mind a reported limitation of the Wnt1-Cre line which has been suggested to not be active early enough to fully explore gene function in EMT in mice (Barriga et al 2015) so perhaps this is why the authors observe more survival and differentiation defects in their mutants.

Overall however, the manuscript contains a lot of novel interesting data about the requirement for proper epigenetic regulation during neural crest cell development, particulalry with respect to craniofacial and cardiac development.

The remaining comments I have are minor:

The Pax3 and Wnt1 Cre lines are described as being active in neural crest cells. This is an oversimplification and technically not correct. For example, Wnt1-Cre is active in the dorsal neuroepithelium which encompasses the territory from which neural crest cells are derived, but Wnt1 is however not expressed in neural crest cells. Pax3 is a little bit different but it's full expression pattern should be described to help the reader understand the phenotype.

Page 10. The authors note the number of SM22a cells were significantly decreased. No statistics were provided however, hence significant should be avoided. Considerably decreased or visibly decreased would be more accurate in the absence of statistics.

Reviewer #2: Authors have satisfied all previous comments and concerns from this reviewer. No additional comments are noted

Reviewer #3: The authors have significantly revised the manuscript and addressed all of my concerns.

**Have all data underlying the figures and results presented in the manuscript been provided?**

Reviewer #1: Yes

Reviewer #2: Yes

Reviewer #3: Yes

PLOS authors have the option to publish the peer review history of their article (what does this mean?). If published, this will include your full peer review and any attached files.

Reviewer #1: No

Reviewer #2: No

Reviewer #3: No

---

## [Editor Report · Decision Letter 2]

25 Feb 2021

Dear Dr Singh,

We are pleased to inform you that your manuscript entitled "Critical role of the BAF chromatin remodeling complex during murine neural crest development" has been editorially accepted for publication in PLOS Genetics. Congratulations!

Yours sincerely,

Frank L Conlon

Associate Editor

PLOS Genetics

Scott Williams

Section Editor: Natural Variation

PLOS Genetics

Comments from the reviewers (if applicable):

**Data Deposition**

http://datadryad.org/submit?journalID=pgenetics&manu=PGENETICS-D-20-01656R2

**Press Queries**

---

## [Editor Report · Acceptance letter]

18 Mar 2021

PGENETICS-D-20-01656R2 

Critical role of the BAF chromatin remodeling complex during murine neural crest development 

Dear Dr Singh, 

We are pleased to inform you that your manuscript entitled "Critical role of the BAF chromatin remodeling complex during murine neural crest development" has been formally accepted for publication in PLOS Genetics! Your manuscript is now with our production department and you will be notified of the publication date in due course.

With kind regards,

Katalin Szabo

PLOS Genetics

On behalf of:
